# Assessment of wind-damage relations for Norway using 36 years of daily insurance data

Ashbin Jaison[1,3], Asgeir Sorteberg[1,3], Clio Michel[2], and Øyvind Breivik[1,2]

[1]Geophysical Institute, University of Bergen, Bergen, Norway
[2]Norwegian Meteorological Institute, Bergen, Norway
[3]Bjerknes Centre for Climate Research, Bergen, Norway

**Correspondence:** Ashbin Jaison (ashbin.jaison@uib.no)

**Abstract.** Extreme winds are by far the largest contributor to Norway's insurance claims related to natural hazards. The predictive skills of four different damage functions are assessed for Norway at the municipality and national levels on daily and annual temporal scales using municipality level insurance data and the high-resolution Norwegian hindcast (NORA3) wind speed data for the period 1985-2020. Special attention is given to extreme damaging events and occurrence probabilities of wind speed induced damages. Because of the complex topography of Norway and the resulting high heterogeneity of the population density, the wind speed is weighted with population. The largest per-capita losses and severe damages occur most frequently in the western municipalities of Norway, more exposed to incoming storms from the North Atlantic, whilst there are seldom any large losses further inland. There is no single damage function that outperforms others. However, a good agreement between the observed and estimated losses at municipality and national levels for a combination of damage functions suggests their usability in estimating severe damages associated with windstorms. Furthermore, the damage functions are able to successfully reconstruct the geographical pattern of losses caused by extreme windstorms with a high degree of correlation. From event occurrence probabilities, the present study devises a damage classifier that exhibits some skill at distinguishing between daily damaging and non-damaging events at the municipality level. While large loss events are well captured, the skewness and zero-inflation of the loss data greatly reduces the quality of both the damage functions and the classifier for moderate and weak loss events.

## 1 Introduction

Wind-related damage claims account for 56% of Norway's insurance payouts related to natural hazards from 1980 to 2017 and are by far the largest component of loss claims related to natural hazards (DSB Norway, 2019). They can affect all sectors from forests to marine and housing infrastructures (Jensen et al., 2010; Gardiner et al., 2013). However, a detailed investigation into the relationship between Norwegian windstorms and damage has so far not been conducted for Norway. The comparison of different proposed storm damage models has only been conducted in a few countries due to a lack of long and sufficiently homogeneous insurance claims data (Cole et al., 2010; Prahl et al., 2015). Determining the best storm-damage model is important in order to make accurate predictions of future damages, whether it be in a few days (short-term forecast) or in many years (climate change range). In this paper, we investigate the relations between windstorms and their associated

damage by analysing 36 years of daily insurance data on the municipality level and daily maximum wind speed data using a set of storm-damage functions. Furthermore, we develop a probabilistic damage classifier that distinguishes between damaging and non-damaging wind speeds to help improve early warning systems.

Establishing robust windstorm-damage relations may help predict storm damage risk in the weather forecasting context (Merz et al., 2020), roughly estimate the storm impact directly after it occurred in order to better plan the emergency response (Welker et al., 2021), and evaluate the change in risk on the longer term in conjunction with climate change. Moreover, understanding the monetary risk involved in extreme wind events is crucial from the insurer's perspective to fix reasonable premiums and estimate portfolio risk. Several methods in the literature assess the risk associated with extreme wind events across various sectors such as agriculture, transport, and energy at varying spatial resolutions (Gliksman et al., 2023). Storm-damage functions are one such method which describes the mathematical relation between the intensity of a natural hazard, here the wind speed, and the monetary loss incurred due to the event. There are mainly two types of storm-damage functions: 1) the storm-based approach, which links historical losses to wind speed information and 2) the hazard-based approach, which in addition makes use of micro-scale information such as the vulnerability, exposure and value of the assets. However, as detailed information about the damage is often proprietary, the most common approach, also used in the present study, is the storm-based approach (Dorland et al., 1999; Klawa and Ulbrich, 2003; Prahl et al., 2012, 2015). These storm-damage functions can also be split into deterministic and probabilistic types (Prahl et al., 2012). The deterministic damage functions do not estimate the uncertainty in the wind speed-loss relation, whereas the probabilistic damage functions assume a statistical distribution for the model error. To make the deterministic and probabilistic models comparable, estimates from deterministic models are treated as equivalent to the mean of the estimates from the probabilistic models.

Storm-damage functions must be regionally fitted because they are highly dependent on local features. The number and spatial extent of the damages caused by extreme wind strongly depend on the exposure level of assets (Cardona et al., 2012), which is connected to demography and economy both changing over time due to a variety of reasons such as urbanisation, higher infrastructure standards, economic growth, etc. Moreover, building types, building codes, differing insurance policies, claims settlement practices can also influence the performance of storm-damage functions (Walker, 2011) especially if they are not locally fitted. Norway has a complex topography with mountains and valleys and a rugged coastline with many fjords with a major share of the population living along the coasts and in the valleys (Simensen et al., 2021). Therefore, the population density is an important factor to take into account when establishing storm-damage functions (Donat et al., 2011a).

A number of studies have investigated storm damage and risk on residential structures and other insured losses, mainly for Europe and more particularly Germany, using various damage functions and local information. Dorland et al. (1999) suggested a deterministic damage function by which loss increases exponentially with wind speed such that a slight increase in storm intensity can cause a significant increase in economic damage in the Netherlands and northwestern Europe. Meanwhile, analysing annual insurance loss due to windstorms in Germany, Klawa and Ulbrich (2003) advocated a cubic relationship between the deviation in wind speed from its 98th percentile and the loss. Donat et al. (2011b) estimated the losses by fitting the Klawa and Ulbrich (2003) damage function at district level for Germany. Heneka and Ruck (2008) and Heneka and Hofherr (2011) applied a probabilistic damage function for Germany, which incorporates extreme value theory in conjunction with a

non-linear function. However, this probabilistic damage function requires both claim and loss ratios, which are not common shared data and which we lack for Norway. To estimate the daily and annual losses at the district levels in Germany, Prahl et al. (2012) proposed a power law-based probabilistic damage function where loss is proportional to a power of wind speed. They found out that the exponents range between 8 and 12, thus highlighting the need to fit the functions locally. Welker et al. (2016) simulated the spatial pattern of losses associated with historical windstorms that happened in Switzerland using the asset amount and the vulnerability, the latter depending on the wind gust. The agreement between the simulated loss and the observed insurance loss was shown to be reasonable but also case-dependent. They pointed to the uncertainty in the input data, such as in the wind gust but also in the estimation of the assets and vulnerability. More recently, Koks et al. (2020) developed an open-source hazard-based model that uses publicly available hazard, exposure and vulnerability data and the loss estimates can be treated as a baseline for further research. Using three different methods, Held et al. (2013) found a steady increase in the values associated with a 10-yr return loss by the end of the 21st century considering only the German private houses. Schwierz et al. (2010) suggested that, with climate change and increased storm intensity, Norway can expect a 16% increase in annual losses associated with windstorms. However, a recent study by Severino et al. (2023) indicated a significant decrease in winter storm damage over Norway.

In the following section, we introduce the insurance loss data and NORA3 hindcast wind speed data along with the different storm damage functions. In section 3, the climatology of the extreme winds and damages is presented in addition to the modelling results. We summarise and discuss the results in section 4.

## 2 Data and methods

### 2.1 Insurance loss data

We use daily insurance loss data, composed of the daily accumulated number of claims and monetary loss, from the Norwegian Natural Perils Pool for each of the 356 municipalities constituting Norway. The data span 36 years, from 1985 to 2020. The loss data distinguish losses by natural event types, such as floods, landslides, storm surges and windstorms. The present study focuses on the damages associated with windstorms.

Natural peril insurance is a compulsory part of the fire insurance held by almost all households in Norway (Sandberg et al., 2020). By the Norwegian Natural Perils Pool act, all buildings and movable properties which are insured against fire damage are also insured against natural disasters. All insurance companies underwriting fire insurance in Norway are obliged to become members of the Norwegian Natural Perils Pool and archive their losses. The fraction of households having fire insurance has stayed relatively constant over the period of interest; thus, the effect of varying market penetrations is small. In many previous studies, loss ratio and claim ratio, which are dimensionless, are used to model storm damage relations (Huang et al., 2001; Held et al., 2013; Prahl et al., 2015; Welker et al., 2016). However, Norwegian insurance does not include the total insured value, which prevents us from using the loss and claim ratios in the present study.

For long time series of loss data, it is necessary to account for inflation. To adjust for the effect of inflation, the insurance loss is modified using the Norwegian official consumer price index (CPI) at a fixed year (SSB Norway, 2023a). The base year

for CPI in Norway is 2015, which we also use here. As an example, the relative difference of loss after inflation adjustment is of +60% for the New Year Storm (1992) and +7% Dagmar (2011).

Changes in exposure is another key variable that determines the intensity of losses incurred. Many studies use population as a proxy for exposure (see e.g., Simpson et al., 2014). Statistics Norway publishes yearly population data at municipality level which goes back to 1951 (SSB Norway, 2023b). To address the change in exposure to a certain extent, we compute the loss per person for each municipality by dividing the insurance loss data with the yearly population. Other factors, which may influence exposure, such as changing building standards and wealth distribution, are not accounted for in the present study.

A few extreme events have caused the majority of the total damage associated with windstorms. The five largest events were responsible for 4.3 bn. Norwegian Kroner (NOK) of damages (2015 values), which represents 36% of the total insurance loss from 1985 to 2020. The top damaging events and their associated losses are given in Table S1. As expected from the more intense cyclones reaching Norway in winter than in summer (Hoskins and Hodges, 2019), most extreme events occur between November and April. The presence of such extreme events brings skewness in the loss distribution and the absence of losses on most days of the year makes loss data zero-inflated (excess number of zeroes in data). The distribution skewness and the zero inflation (Fig. S1) in loss data is challenging for conventional fitting methods, such as the least-squares or maximum likelihood. Figure S1 highlights a record high number of claims in years 1994, 2015, and in a lesser extent 2011. This can be attributed to the storm of 1994, the storm Dagmar in 2011 and the storms Nina and Ole in 2015 (Table S1).

## 2.2 The wind speed data from the NORA3 hindcast

The high-resolution hindcast NORA3 covers the period 1979-2021 (and is being extended). The spatial resolution of NORA3 is 3 km×3 km, and surface variables are archived at hourly resolution. The non-hydrostatic, convection-permitting model HARMONIE-AROME (Seity et al., 2011; Haakenstad et al., 2021; Haakenstad and Breivik, 2022) ingests surface observations through a simple surface analysis scheme and blends this with initial fields from ERA5 (Hersbach et al., 2020). Boundary conditions are also taken from ERA5. The data is publicly available on the website https://thredds.met.no/thredds/catalog/nora3/catalog.html (last access 01 October 2023). The domain covers the North Sea, the Norwegian Sea, the Barents Sea, Svalbard and is bounded by Finland to the east. The hindcast consists in a sequence of 9-h forecasts initialised at 00, 06, 12 and 18 UTC every day from 1985 to 2020, which were the 36 years available at the time of our analysis. Aggregating the 4-9 h lead times provides an hourly dataset from which we extract the daily maximum wind speed and gust. NORA3 only slightly underestimates the maximum observed wind speed (Haakenstad et al., 2021; Solbrekke et al., 2021) and its interquartile range for the 10 strongest windstorms that affected Norway between 2009 and 2018 (Haakenstad et al., 2021), outperforming both the earlier hydrostatic 10-km Norwegian Hindcast Archive (NORA10, Reistad et al., 2011) as well as the recent ERA5 reanalysis.

## 2.3 Municipality level wind speeds

As the insurance loss is at the municipality level, we must estimate a municipality-relevant wind speed to apply the storm-damage functions. A simple approach is to average the daily maximum wind speed over all grid points contained in a given municipality. However, to compensate for the complex topography and disparate demography of Norway, we calculate a

population-weighted wind speed to remove extreme wind events occurring over mountains, lakes, and other population-sparse regions. We make use of the gridded population data at 1 km×1 km for Norway for the period 2001 to 2019 (https://www.ssb.no/natur-og-miljo/geodata; Strand and Bloch (2009)). As it does not cover the whole period of the study, we compute the average of population in each grid cell over the available period (2001-2019). Then this averaged population is remapped on the same 3 km×3 km grid as the NORA3 data. To achieve this, we assign each population grid cell to the nearest NORA3 grid cell. If more than one non-zero population grid cell corresponds to a NORA3 cell, we assign the sum of the population grid cells to the NORA3 grid cell. Finally, in order to have the wind speed at the municipality level, as is the insurance data, we take the population-weighted average of the daily maximum wind speed in each municipality. We repeat the process for the daily maximum wind gusts.

## 2.4 Storm-loss models

Storm-damage functions connect the intensity of a storm event to the monetary damage caused by the storm. With the available historical data of insurance loss and wind speed, we apply the storm-based approach to fit several storm-damage functions. The storm-damage functions discussed here are macroscale statistical models calibrated at the municipality level. Our key objective is to compare and assess the quality of various proposed storm-damage functions applied to our data. We employ three damage functions: the deterministic exponential model (Dorland et al., 1999; Huang et al., 2001; Murnane and Elsner, 2012), the deterministic model of Klawa and Ulbrich (2003) and the probabilistic function by Prahl et al. (2012). In addition, we suggest a modified version of the Prahl model to better simulate the very steep damage curves found in some Norwegian municipalities. All damage models are fitted to loss per person to ensure uniformity among the storm damage approaches and easier inter-comparison of models and parameters. Finally, we devise a simple ensemble mean of the estimates from the four damage functions listed above, to check if it performs better than any of the four individual functions. In the following, we describe in detail the damage functions applied. From now on, $L$ refers to the insurance loss, $\nu$ to the weighted wind speeds and $d$ to the damage function.

To fit and assess the skill of the storm damage models, we split the data into a testing and a training set. We assign the years from 1985-1989 and 2010-2012 to the testing part. The rest of the data from 1990-2009 and 2013-2020 is the training data. A necessary condition for splitting the data is that training and testing data should have identical distributions. We split the data so that both testing and training data include extreme storm events, the storm Dagmar in the testing data, and the New Year storm in training data.

For robust storm-damage relations, extreme care should be taken while calibrating the damage functions. To make sure that the small losses, which are more frequent, are not better fitted than the high losses which are much less frequent, we bin the loss data with respect to wind speeds to reduce the weight of low loss events. Note that we do not perform binning for the Klawa damage function as the model is only suitable for high loss events and inherently decreases the number of zero and low losses with the use of a high wind speed threshold. More about binning in individual models is explained in the following sections.

As many previous studies before (Donat et al., 2011b; Prahl et al., 2012, 2015; Pardowitz et al., 2016), we choose to fit the storm-damage functions at the municipality level. Despite the issue with a larger number of zero or low losses, this method

is more meaningful and has the potential to be more accurate than doing fits using a country-averaged population-weighted wind speed. For example, the number of loss days when the population weighted wind speed exceeds its 98th percentile is high along the Norwegian coast and low for further inland regions (Fig. S2a). Moreover, there are many local factors influencing the damage, such as building rules, building types, that promote local fits of the storm-damage functions. As a test, we pooled all municipalities together to perform the fits, but it didn't show any improvement in the estimation of both municipal and national level losses (Fig. S3).

### 2.4.1 Exponential model

The exponential damage function assumes that loss increases exponentially with increasing wind speed (Dorland et al., 1999). It is a simple damage function with only two parameters to be estimated and is formulated as,

$$d(\nu) = e^{\alpha(\nu - \beta)} \tag{1}$$

where $\alpha$ is the scale parameter and $\beta$ is the location parameter. The loss is estimated from the damage function as $L(\nu) = d(\nu)$. The exponential model, by its shape, can be extended to low wind speeds that may cause low to medium size losses. To take advantage of this, we choose the 95th percentile of the population weighted wind speed in each municipality as the threshold for the exponential model above which the aggregated losses represent 82% of the national losses that occurred in the training period. Such a threshold ensures that the model accounts for low to medium losses while discarding the very small losses in the lower loss spectrum. The associated loss values are split into ten equally spaced bins with respect to the wind speeds and with a pre-condition that at least five loss days belong to each bin, as in Prahl et al. (2015). Note that Fig. 1a only displays 6 bins because the 4 other bins do not include the minimum of 5 loss days required in each bin. The binned losses are log-transformed, and with the assumption of normality, least square method are employed to estimate the model parameters. Figure 1a shows the shape of the damage function with the red line. Although we only use the wind speed bins above the 95th percentile of the wind speed to calculate the fit, the obtained exponential model can also be applied to the wind speeds below the 95th percentile and we can get loss estimates for those wind speeds as well, as shown in Fig. 1a (see the red dashed line).

### 2.4.2 Cubic-excess over threshold model

The damage function proposed by Klawa and Ulbrich (2003) suggests that the loss increases cubically for wind speeds beyond a certain threshold. The Klawa model was originally developed as a loss index for Germany to estimate annual national losses using the German insurance data. Later, using the same insurance data, the damage function was calibrated by Donat et al. (2011b) for the German districts and by Pinto et al. (2012) for the affected areas of individual storm events. In the present study, we chose to calibrate the Klawa damage damage function with insurance loss at municipality level similar to Prahl et al. (2015) who applied it at district level on daily German insurance losses. This damage function takes the third power of wind speeds above the 98th percentile of the wind speed determined using the whole study period (1980-2020) scaled by the same 98th percentile of the wind speed:

$$d(\nu) = \left( \frac{\nu - \nu_{98}}{\nu_{98}} \right)^3 \tag{2}$$

The loss is obtained by linear regressing the damage function:

$$L(\nu) = \beta_0 + \beta_1 d(\nu) \tag{3}$$

The intercept term $\beta_0$ in the fitted linear regression can be interpreted as the base loss, which is the loss estimate for all wind speeds below the 98th percentile. However, using this loss offset for all wind speeds below the 98th percentile doesn't allow to address the randomness in the lower loss spectrum. Figure 1b shows the model fit for this damage function (see the red solid and dashed lines). $\beta_1$ is the slope of the line. The two $\beta$ parameters are obtained using a least-squares regression method.

Several studies across Europe used the 98th percentile wind speed as a threshold for the Klawa damage function (Pinto et al., 2012; Karremann et al., 2014a, b). Ideally, the threshold for damaging wind should be locally chosen using statistically-determined estimates, but for simplicity we have kept the often used 98th percentile. In Norway, 72% of the insured losses are caused by wind speed above the 98th percentile. As the Klawa model is not designed for low loss cases, this is a fairly reasonable simplification. Note that if grid point wind speeds are chosen, this choice of percentile can be problematic for places with weak winds, such as southeastern Norway (see Fig. S4a).

To alleviate this, Karremann et al. (2014b) and Little et al. (2023) suggested a 9 m/s fixed threshold for wind speed causing damage in Norway. However, in our study, we do not need this 9 m/s threshold as we use the population-weighted averaged wind speeds, reducing the relative importance of grid cells with very low wind speeds and therefore avoiding the problem of very low 98th percentile. Note that even wind speeds above the 98th percentile can be associated with no loss. Figure S2a shows that this often happens in southern inland regions of Norway, where it contributes to the uncertainty in the loss estimation.

Here we weight the wind speeds with population and aggregate it to the municipality-level resolution such that it corresponds to the loss data resolution. However, other studies, such as Pinto et al. (2007), weight the loss index and aggregate it to the district or national resolutions. As discussed later in the paper, these two methods do not give very different results.

### 2.4.3 Probabilistic damage function by Prahl

The power law based probabilistic damage function by Prahl et al. (2012) consists of a two-step fitting procedure: the first step estimating the occurrence probability of damage for a given wind speed and the second step estimating the loss magnitude. For both steps, we use wind speed bins and each bin must have at least five loss days, as (Prahl et al., 2015) did. We then fit the following sigmoid function to the binned wind speeds.

$$p(\nu) = 1 - \frac{\gamma_0}{1 + e^{\gamma_1 (\nu - \gamma_2)}} \tag{4}$$

where the parameter $\gamma_1$ determines the steepness of the curve, $\gamma_2$ is the wind speed threshold beyond which the curve gets steeper and $\gamma_0$ determines the base probability of losses. Figure 1e shows the fit of the probability term (Eq. 4) of the damage

function (see the red line). In addition, for a given wind speed $\nu$, the magnitude of the loss M for non-zero losses is estimated through a power law-based function (Fig. 1c) and is related to the wind speed as follows:

$$M(\nu) = \sigma_0 + \left(\frac{\nu}{\sigma_2}\right)^{\sigma_1} \tag{5}$$

where $\sigma_2$ scales the wind speed, $\sigma_1$ is the shape parameter and $\sigma_0$ is the offset loss. The magnitude term is fitted on losses binned with respect to wind speeds.

The probability term makes use of the whole loss range while the magnitude term only uses non-zero losses. The probability of damage and the magnitude of loss are treated as independent variables. The damage function is then the product of the probability and the magnitude of loss:

$$d(\nu) = p(\nu)M(\nu) \tag{6}$$

    The damage function includes the assumption that the observed losses follow a log-normal distribution ($M_{obs} \sim \mathrm{LN}(\mu, \sigma)$,
where $M_{obs}$ is the observed non-zero loss). Therefore, the expected loss for a given wind speed is

$$L = p(\nu)E(M(\nu)) \tag{7}$$

The probabilistic damage function by Prahl has a complex fitting procedure with eight parameters to be estimated. We refer the readers to the work by Prahl et al. (2015) to learn more about the parameters and fitting procedures of the model. The location ($\mu$) and scale parameters ($\sigma$) of the log-normal distribution are estimated using the maximum likelihood method, and the other
parameters of the damage function are estimated with the least squares method.

### 2.4.4   Modified probabilistic damage function by Prahl

The rationale behind Prahl's damage function is that the loss increases steeply for extreme wind events (Fig. 1c). However, based on inspection of the quality of the fitted curves for very high loss events, we identified a need for an even steeper damage function for certain municipalities in Norway. As the deterministic exponential damage function increases sharply and shows
good fits for some of the municipalities, we propose a modified version of the damage function by Prahl that combines an exponential fit with the probabilistic aspect of the Prahl model. The magnitude term in Eq. (5) of the Prahl damage function is modified as follows:

$$M(\nu) = \sigma_0' \exp\left[\left(\frac{\nu}{\sigma_2'}\right)^{\sigma_1'}\right] \tag{8}$$

The rest of the fitting procedure and assumptions are the same as for the Prahl damage function. The shape of the magnitude
term in Eq. (8) is displayed in Fig. 1d with the red line.

### 2.4.5 Ensemble mean method

The four damage functions presented above have different advantages and drawbacks. The ensemble mean is calculated as the arithmetic mean of the loss estimates of the four functions, in the hope to improve the overall accuracy, as proven for ensembles of numerical weather/climate simulations.

### 2.5 Damage classifier

A damage classifier labels a given wind speed as damaging or not and adds useful information for event preparedness. The probabilistic damage occurrence probability function in Eq. (4) gives us the opportunity to define a classifier that distinguishes between a damaging and a non-damaging event. To build a robust classifier, it is necessary to define the probability threshold that separates an event from a non-event. With non-event days outnumbering the event days (class imbalance), it is not straight-forward to define the probability threshold as 0.5 or to evaluate the model performance for various probability thresholds on the basis of traditional receiver operating characteristic (ROC) curves and the corresponding area under the curves (AUC). To circumvent the problem of class imbalance in identifying the best probability threshold, we employ the precision-recall curve and the associated F-scores (cf. section 2.6, Sokolova and Lapalme (2009)). Figure 2 shows the precision-recall curve for an arbitrary municipality. The split point of damage classifier corresponds to the probability threshold with the highest F-score from the precision-recall curve.

### 2.6 Model evaluation metrics

We evaluate the models' performance on the training and testing parts at the municipality level using the mean absolute error (MAE) and coefficient of variation (CV). In addition, the predictive skill of the probabilistic function in the Prahl damage function is evaluated using accuracy, recall, precision and F-scores.

As its name indicates, MAE is the mean of absolute differences between the observations and the model fits and is formulated as:

$$\text{MAE} = \frac{1}{n} \sum_{i=1}^{n} |y_i - \hat{y_i}| \tag{9}$$

where $y_i$ is observed loss and $\hat{y_i}$ is the estimated loss. A high MAE indicates a poor skill of the model. Another evaluation metric used here is CV based on the root mean square error defined by Prahl et al. (2015) as follows,

$$\text{CV} = \frac{1}{\bar{y}} \left( \frac{1}{n} \sum_{i=1}^{n} (y_i - \hat{y_i})^2 \right)^{\frac{1}{2}} \tag{10}$$

where $\bar{y}$ is the mean of the observed loss. High values of CV indicate large loss variability compared to the mean loss.

To quantify the classification skill of the damage classifier, we employ the precision, recall, accuracy and F-scores, which are defined as follows:

- Precision: The proportion of correctly classified positive samples to the total number of samples classified as positive.

– Recall: The proportion of correctly classified positive samples to the total number of positive samples.

   – Accuracy: The proportion of correctly classified samples to the total number of samples.

   – F-Score: Theoretically, the F-score is defined as the harmonic mean of precision and recall. It indicates the balance between precision and recall. F-score ranges between 0 and 1 and the higher the F-score the better. We take advantage of the F-scores to define the probability threshold for the damage classifier.

The binary damage classifier is optimized using the precision-recall curve and associated F-scores. The precision-recall curve is obtained by calculating the precision and recall for all potential probability thresholds obtained from the observed occurrence probabilities. The F-scores are computed for all points of the precision-recall curve (i.e. all probability thresholds) and evaluate the ability of each probability threshold to minimize false positives and capture true positives simultaneously. The probability threshold at which the F-score is maximum is chosen as the split point for the event classifier.

The damage functions are sensitive to extreme loss observations and the presence of few extreme events can heavily alter the damage functions' shape. Therefore, different training data sets may result in differing damage function fits. Cross-validation is an effective method to estimate the uncertainties involved in the choice of the testing and training data. We perform a seven fold cross-validation by splitting data into seven with each set of testing data having five consecutive years of data. So, in the first fold the testing period is 1985-1989 and the training period is 1990-2020, in the second fold the testing period is 1990-1994

and the remaining years are in the training dataset, and so on.

## 3   Results

In this section, we analyse the spatial and temporal distribution of the insurance loss and compare the population weighted daily maximum wind speed, population weighted daily maximum wind gust and the daily maximum wind speed at the municipality level. We then compare the different damage functions along with the modified Prahl and ensemble mean models. However,

considering the high degree of detail involved, we emphasize the following aspects, (1) daily losses at the municipality level, (2) top three extreme damaging wind events during the study period, (3) losses aggregated to the national level and (4) the probability function in Prahl et al. (2012) as a classifier. Furthermore, we discuss the pitfalls of the loss data, wind data and storm damage functions.

### 3.1   Overview of the windstorms losses

The municipalities on the west coast of Norway experience higher losses per person whereas there is hardly any loss further inland in southeastern Norway (Fig. S5a). Skewness and zero inflation are especially high for some municipalities in southeastern Norway where wind-related losses are rare. This rarity of loss days greatly limits the performance of the damage functions.

    The ten most damaging windstorms, in terms of cost, that reached Norway during the study period occurred between October and March and mainly affected central and southwestern Norway and more marginally Northern Norway (Table S1). An

305 example of such a damaging storm is Dagmar in 2011 that affected western Norway causing more than one billion NOK of

losses (Fig. S5b). The insurance losses caused by the ten largest events are given in Table S1 and represent a total of 5347 million NOK, which is 44% of the total losses due to windstorms between 1985 and 2020.

We find no significant temporal trends in the insurance losses caused by extreme winds. Trends in the losses time series, should arise from inflation or changes in wealth distribution. However, the effect of inflation is nullified by adjusting the insurance losses with the consumer price index and a change in wealth distribution is overlooked by the skewness in the losses. Therefore, the Mann-Kendall trend test we conducted on the annual national losses (Fig. S6) fails to detect any significant trend in losses.

The choice of wind data has the potential to influence the performance of the damage functions (Prahl et al., 2015). Also, the 98th percentile wind speed is widely regarded as critical from a damage perspective (Klawa and Ulbrich, 2003; Schwierz et al., 2010; Donat et al., 2011a). Figure S4a shows that the west coast and Northern Norway experience high magnitude wind events in comparison with southeastern municipalities. The 98th percentile of the population-weighted daily maximum wind speed exhibits a high correlation with the 98th percentile of the population-weighted daily wind gust (0.91) but a lower correlation with the 98th percentile of the unweighted daily maximum wind speed (0.61) (Fig. S4b). This difference can be attributed to the added information of population as weights for wind speed and emphasizes the importance of accounting for demography.

From the damage perspective, extreme damaging events are of the topmost concern. For each municipality, we define the losses higher than the 99.7th percentile as the extreme loss class and losses lying between the 98th and 99.7th percentiles as the high loss class. The aggregated municipality losses in the extreme loss class account for 85% of the total national loss, while the high loss class comprises 8% of the total national loss. In each municipality, the extreme loss class includes approximately 31 days in the training data and 9 days in the testing data (occurring on average around once a year). Segregation of losses into different classes helps to assess the performance of the damage functions for events of different severity.

By applying the different damage functions, we get daily fits of insurance loss for 10227 days in the training dataset and predictions for 2922 days in the test dataset for every municipality in Norway.

## 3.2 Municipality level loss estimations

To demonstrate the advantage of weighting wind speed with population, damage functions were also fitted with the original wind speed as the predictor variable. The prediction error on the test data shows that the population weighted wind speed has lower CV in 67% of municipalities (see also Fig. S4c in supplement for the spatial distribution of where the original wind speed data performs better). From these results, we conclude that weighting wind speeds with population tends to improve the predictive performance of the damage functions. Therefore, from now on, we only use the population-weighted wind speeds when fitting the damage functions. The deterministic damage models, that are the Klawa and exponential damage functions, perform best in nearly two thirds of the municipalities across all losses classes in terms of MAE. Table 1 shows the performance of the four different damage functions defined in the methods section and of the ensemble mean for different loss classes. The deterministic models exhibit the smallest median MAE across all municipalities. Using the CV as the evaluation metric gives similar models performances as when using the MAE. The ensemble mean method does not massively outperform the competing models, but tends to give better results than the two Prahl's damage functions (Table 1). A map of the best model

for each municipality exhibits a high heterogeneity with no obvious spatial pattern, that is no model performing best in certain regions (Fig. S2b). Overall, our results suggest that the Klawa storm-damage function is the best model for a large share of municipalities (37.6%).

      The spatial distribution of MAE is not uniform, but can be linked to the magnitude of the variance of losses, with municipalities with large loss variance having the largest MAEs (Fig. 3). In addition, the spatially heterogeneous distribution of losses

(Fig. S5a) gives rise to spatially heterogeneous errors (Fig. 3b). The CV, which shows the extent of variability in losses in relation to the mean losses, exhibits a relatively heterogenous structure (Fig. 3c). On the one hand, there is tendency for high CV in some inland municipalities of southeastern Norway, where the rarity of windstorms could be part of the reason. On the other hand, the northwest part of southern Norway also exhibits high CV although windstorms are more frequent there (Fig. 3b).

Pooling all municipality-level population-weighted wind speeds together to perform the storm-damage functions fits does not give better municipality-level losses estimations, as expected because of local effects, such as e.g. different population density and vulnerability. The most skilled model in each municipality is associated with a larger MAE than when the fit is performed at municipality level (see Fig. S3a). This reduction of skill also occurs for the national-level losses (Fig. S3b).

      Unlike previous studies (e.g., Pinto et al., 2007), which weighted the spatially aggregated loss index devised from cubic

exceedance of wind speed above a sufficiently high threshold (computed as in eq. 2), we here weight the wind speed with population first and then aggregate it to a coarser resolution. We compare the Klawa damage function as in eq. 2 obtained from the proposed methodology with the alternative methodology employed in Pinto et al. (2007). We found both the damage estimates and its error to be strongly correlated (Fig. S7a). An independent sample t-test failed to conclude for any significant differences between the mean of MAEs from both methodologies. A detailed comparison can be found in the supplemental

material.

      The seven fold cross-validation reveals that the parameters in the storm-damage functions obtained during the fitting step depend on the choice of the training dataset. Moreover, the model evaluation metrics are highly dependent on the choice of the training dataset (see the range in Fig. S8 a,b,d,e,g,h). However, independent of the training dataset, the Klawa and exponential models have the best skill in most of the municipalities (as also shown in Table 1) across the different loss classes (see Fig. S8

c,f,i).

      The fits of the four damage functions and of the probability term largely vary not only between models but also from municipality to municipality (Fig. S9). Figure S9a,c,d illustrates the variety among the fits for the exponential, Prahl and modified Prahl damage functions with very steep lines for some municipalities and much flatter lines for others. Figure S9b highlights that the Klawa damage function does not increase as steeply as the other models and the variability among the

municipalities is smaller. Finally, Fig. S9e also shows that the sigmoids depicting the probability of damage occurrence have different shapes in different municipalities with some curves not reaching a loss probability of 1 within the wind speed range represented. Note that the fit can lead to negative probabilities, that we set to 0 afterwards.

### 3.3 Extreme damaging events

As extremely damaging windstorm events are of foremost importance, for example from the insurers' point of view, the ability of the damage functions to reproduce the damages associated with these events has to be assessed. To compare the estimated and observed losses caused by major storm events, we sum the loss within the date range as given by the Norwegian Natural Perils Pool in Table S1. Using only the model exhibiting the best performance, on the whole testing period, in each municipality, we manage to reproduce the spatial pattern of the damages for the three most damaging wind storm events (Fig. 4, see Fig. S10 for estimates from individual models for the three most damaging wind storm events and Table S2 for their corresponding correlations. Also, Fig. S11 shows spatial patterns of seven other damaging events as given in Table S1.). Statistically significant spatial correlations between the observed and estimated losses reaffirm the suitability of the damage functions to estimate the economic impacts of extreme damaging events.

In the extreme loss class, the probabilistic damage functions and the Klawa damage function perform best in a third of the municipalities each. The Klawa damage function also shows the smallest median error in the extreme loss class which is in agreement with previous comparison studies on storm damage functions over Germany (Prahl et al., 2015).

### 3.4 National level loss

Aggregating the municipality level loss observations and estimates yields a time series of daily national loss for each model and we find an overestimation of low-magnitude losses as all damage functions are calibrated in favour of extreme losses (Fig. 5). Moreover, the models' estimates capture well the magnitude and temporal evolution of the observed annual losses at the national level, with a Spearman rank correlation of 0.84 (Fig. 6). Figure 6 also reveals that the losses in the extreme loss class are slightly overestimated in the training period in years where extreme storm events have occurred while there is an underestimation of loss in 2011 (part of testing data) when storm Dagmar occurred. The aggregated annual national level losses for individual models are shown in Fig. S12 and Fig. S13. Figure S12 shows that the deterministic models are well able to estimate losses in the extreme loss class at the national level. The probabilistic models overestimates the losses in certain municipalities by a large margin reducing the models' ability to estimate national level losses (Fig. S13). While fitting the probabilistic damage functions, there are not enough extreme loss observations in certain municipalities, which prevents us from requiring a minimum number of loss observations in each bin. This is one of the reasons for the very large differences between the observed and estimated losses for the probabilistic models.

### 3.5 Probability of damage occurrence

The damage classifier, devised here from the probability term in the Prahl's function (see section 2.6), demonstrates some skill at predicting the most extreme events, but struggles for the weaker events. It correctly predicts the top five extreme events (Table S3) for over 70% of the municipalities. Even though we try to address the excess number of zeroes with the precision-recall curve, the classifier was only able to detect 20-40% of the actual damaging events in most municipalities (see the number of municipalities in the [0.2-0.4] interval in Fig. S14a) and zero events in around 15% of the municipalities ($\simeq$ 50 municipalities

for the null true positive rate in Fig. S14a). Moreover, the false positive rate is small ($< 4\%$) in all municipalities (Fig. S14b). Because of the noisy lower loss regime, the calculated probability thresholds are low (Fig. S15) for most municipalities with values between 0.02 and 0.4 (median of 0.23, Fig. S15a). Only a few municipalities exhibit probability thresholds above 0.4, especially in southeastern Norway where damaging wind events are rare (Fig. S15b).

Although its skills are relatively poor, the damage classifier defined from event occurrence probabilities clearly outperforms
a classifier that solely relies on wind speed. To demonstrate this, we define a damage classifier based on wind speed thresholds in which all wind speeds above the 98th percentile are labelled as damaging (as is done in the Klawa model). A comparison between these two classifiers shows far higher accuracy for the classifier using the probability threshold (Fig. 7b) than for the classifier using the wind speed classifier threshold (Fig. 7a).

## 4  Conclusions

Windstorms are the natural hazard that makes up more than half of the monetary losses in Norway. The capability of four storm damage functions and their mean to reproduce the monetary losses associated with damaging wind events is evaluated for the complex topography and demography of Norway. The models' ability to reproduce spatial loss patterns of extreme loss events with a high degree of accuracy confirms the utility of both deterministic and probabilistic damage functions in estimating extreme loss events. However, the relatively poor performance of the damage/no-damage classifier points towards
the difficulty of developing an early warning system that encompasses also small loss events. Our results confirm the importance of weighting wind speed with population, of locally fitting the storm-damage functions and of using various damage functions to best estimate the windstorms losses.

The deterministic Klawa model performs best in estimating extreme losses and this result is consistent with previous studies, such as Prahl et al. (2015). In our study, the Klawa model also exhibits the smallest error in the entire loss range. But, the Klawa
model's inability to account for losses associated with wind speed below the 98th percentile greatly limits its applicability in the lower loss range. The Prahl damage functions have the ability to model the whole loss range and show the smallest error in a third of the municipalities. The models' performances suggest that relying on one single damage model may not be the best strategy if all the municipalities in Norway are to be modelled. Due to the high spread in the fits of damage functions, the ensemble mean method mostly fails to outperform the individual models. Although the damage/no-damage classifier does very
well at predicting extreme damaging events, more research is needed to propose a well-functioning damage classifier across all loss ranges.

Wind speed is the most common variable used to estimate storm damages. A drawback of this approach is that the same wind speed at the municipality level resolution may cause small damages in some cases or no damages in most cases. Such inconsistencies occur mainly due to extremely local high wind gusts and incorrect reporting of damages. As a consequence, the
lower end of the wind speed-damage relations becomes noisy, thus making it very difficult to model. To check how the wind gust from NORA3 compares to the wind speed, we performed the same population weighting exercise with daily maximum wind gusts and found a high correlation (0.91) in the 98th percentiles calculated from wind speeds and wind gusts (Fig. S4b).

With insurance data being at a coarser resolution than the wind gusts, which are very local (a few hundred metres), it is difficult to get meaningful wind gust values at municipality level because the impact of high values will be weakened by the population-weighted averaging step.

Due to the unavailability of the gridded population data for the earlier part (1985-1999) of our study period, we had to use a constant spatial distribution of the population to weight the wind speed at every grid point. Therefore, we cannot take into account the spatial change in population density, such as the spatial expansion of cities with time. This is a source of uncertainty in our storm-damage fits.

High quality data on loss and wind speed is necessary for the calibration of damage functions. Long time series of loss data are desired to reduce uncertainties and increase accuracy of model fitting and predictions. However, loss information as used in this study is rarely available. In such cases, a general approach is to approximate the losses using population of the respective regions and then quantify the impact of windstorms (Donat et al., 2011a). In addition, there are open source climate risk assessment models such as CLIMADA (Aznar-Siguan and Bresch, 2019), which can be coupled with loss data for damage estimation.

There are several limitations to the damage functions including the inability of the models to account for the duration of the events and their tuning to model extreme losses at the expense of the low losses. Furthermore, the randomness of losses towards the lower loss spectrum diminishes the damage classifier's predictive skill. There are also certain pitfalls in the insurance data, such as incorrect reporting of time, location and type of claims. Also, the slight underestimation of maximum wind speeds in NORA3 may affect the shape of the damage curves. A direct comparison between other studies that employ damage functions is not possible because the unit of loss in this study is NOK per person while most other studies use the loss ratio (insured loss/total value of the insured assets) instead of the actual insured loss.

Applications of damage functions can range from impact-based forecasting of damage, to damage assessment right after an event, as well as assessment of future losses in the context of climate change with an ensemble of wind-damage relations providing a measure of the uncertainty in the monetary loss amount. Previous studies suggest that with climate change the intensity of future windstorms may increase (see, e.g., Priestley and Catto, 2022; Michel and Sorteberg, 2023). It would be worthwhile to assess the future changes in windstorm-induced losses using the damage functions discussed here and future wind speed projections. Impact-based forecasting by which risks associated with a natural hazard are predicted on the short term is gaining more popularity for climate risk management (Taylor et al., 2018; Zhang et al., 2019). The performance of these damage models, especially on regional level, suggests their utility for impact-based forecasting. However, to use trained storm-damage models on new data, one has to make sure that the distributions of the wind speed in the training dataset and the testing dataset are identical. To ensure this, statistical adjustments methods may be required. For forecasting purposes, an ideal starting point would be to apply a damage classifier to distinguish between damaging and non-damaging winds, as part of an early warning system, followed by a prediction of losses using a variety of damage functions. Also, from the risk modeling perspective, coupling the damage functions with the asset exposure, i.e. information on infrastructures in addition to the population density, is a possible future direction.

*Author contributions.* AJ: conceptualization, analysis, visualisation, interpretation of results, drafting of the paper. AS: conceived the idea, conceptualization, supervision of the work, funding, interpretation of results. CM: post-processing of NORA3 data. ØB: supervision of the work, funding. All authors contributed to drafting and reviewing the manuscript.

*Competing interests.* The authors declare that they have no conflict of interest.

*Acknowledgements.* This work was funded by the Research Council of Norway through the project StormRisk nr 300608 granted to A. Sorteberg. We also thank the Norwegian Natural Perils Pool and the Norwegian Meteorological Institute for providing the data. We are grateful to the two reviewers who gave insightful comments on the manuscript helping to improve the quality of the manuscript.

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

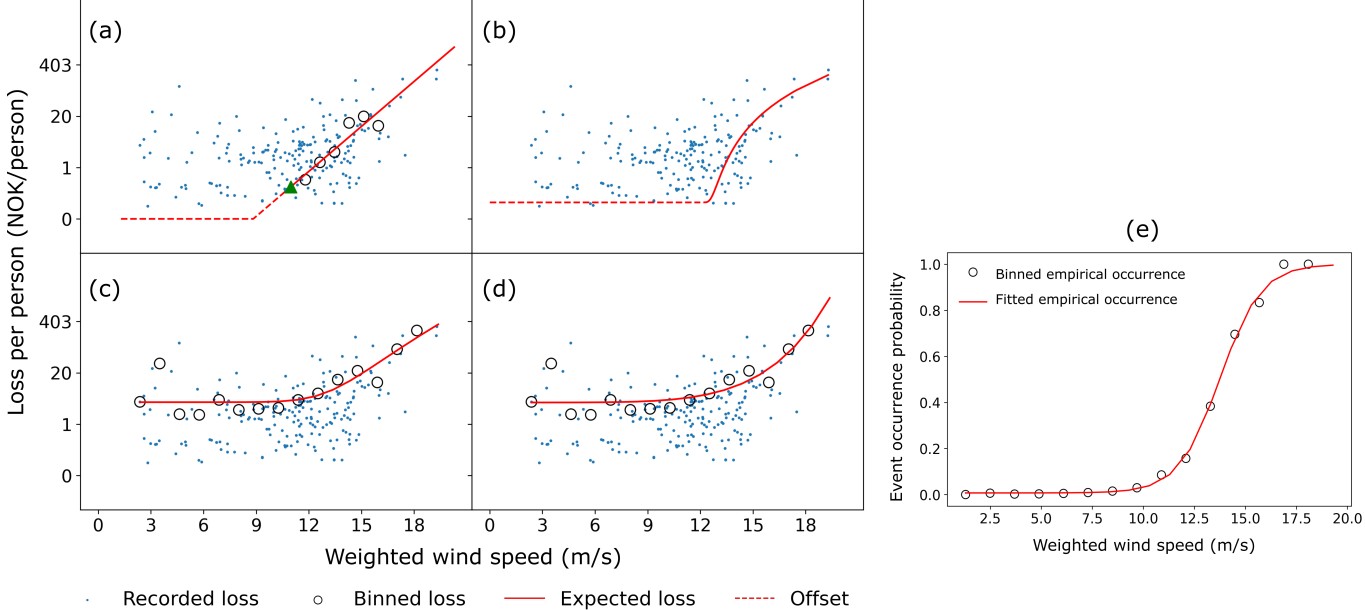

**Figure 1.** Shapes of the damage functions for an arbitrary municipality for (a) the exponential damage function where the green triangle denotes the loss corresponding to the 95th percentile of the wind speed and the red dotted line represents the loss estimates below the 95th percentile of the wind speed, (b) the cubic excess over threshold damage function and the red dotted line represents the loss estimates below the 98th percentile of the wind speed, (c) the magnitude term in the probabilistic damage function by Prahl, and (d) the magnitude term in the modified Prahl probabilistic damage function, (e) example of sigmoid function that estimates the probability of an event occurrence for an arbitrary municipality. The estimated parameters in this municipality are: $\gamma_0$=0.99, $\gamma_1$=0.99 and $\gamma_2$=13.75. Note that the y-axis for (a)-(d) is on a logarithmic scale and the zero loss on the y-axis is only for reference but the zero losses are not plotted/displayed.

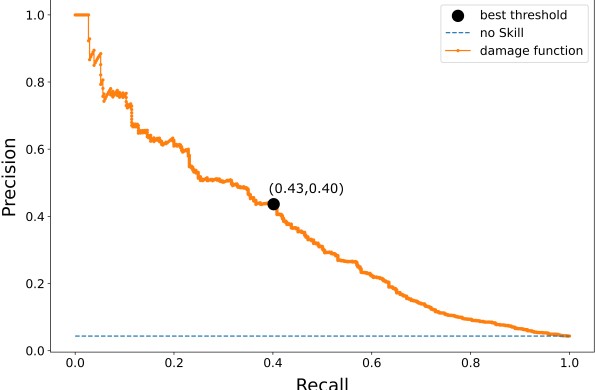

**Figure 2.** The precision-recall curve for an arbitrary municipality is shown with the orange line, the dashed blue line is the model with no skill, and the black dot corresponds to the point where the F-Score is maximum. In this example, the highest F-score of 0.41 is achieved at the probability threshold of 0.30. The precision and recall are shown in brackets.

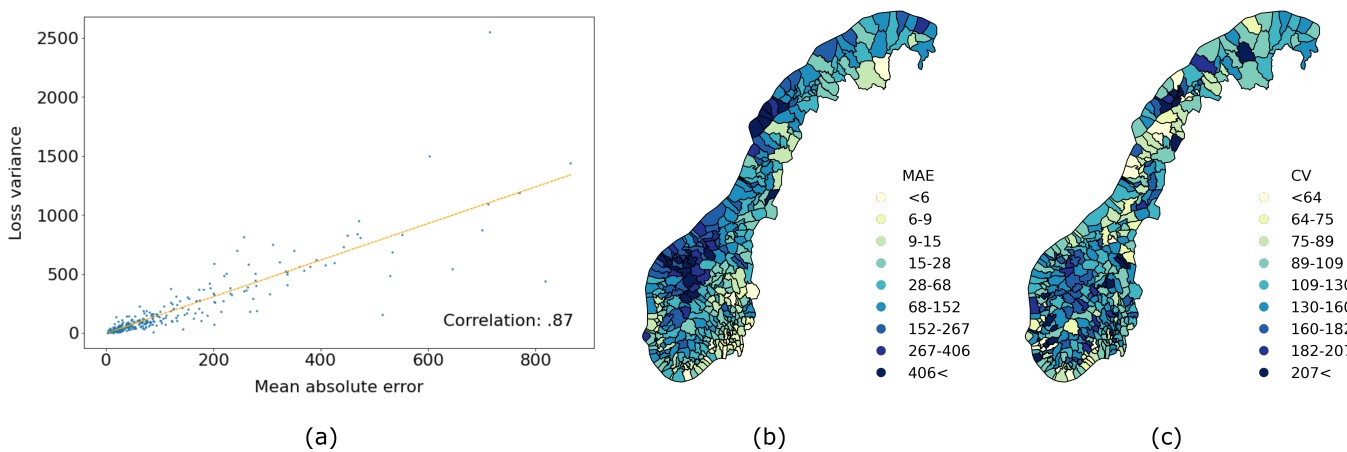

**Figure 3.** (a) Scatter plot of the loss variance against the smallest MAE for losses above the 99.7th percentile in the testing data where each dot represents a municipality. The orange line represents the linear trend obtained using a least squares regression with the correlation indicated in the bottom right corner. (b) Map of the smallest MAE among the five models in the extreme loss class fitted on the test data and (c) the corresponding coefficient of variation of the root mean square error. In (b) and (c), the legends have non-linear class boundaries at the 5th, 10th, 20th, 40th, 60th, 80th, 90th and 95th percentiles. Note that the results are based on the performances on the unseen testing data.

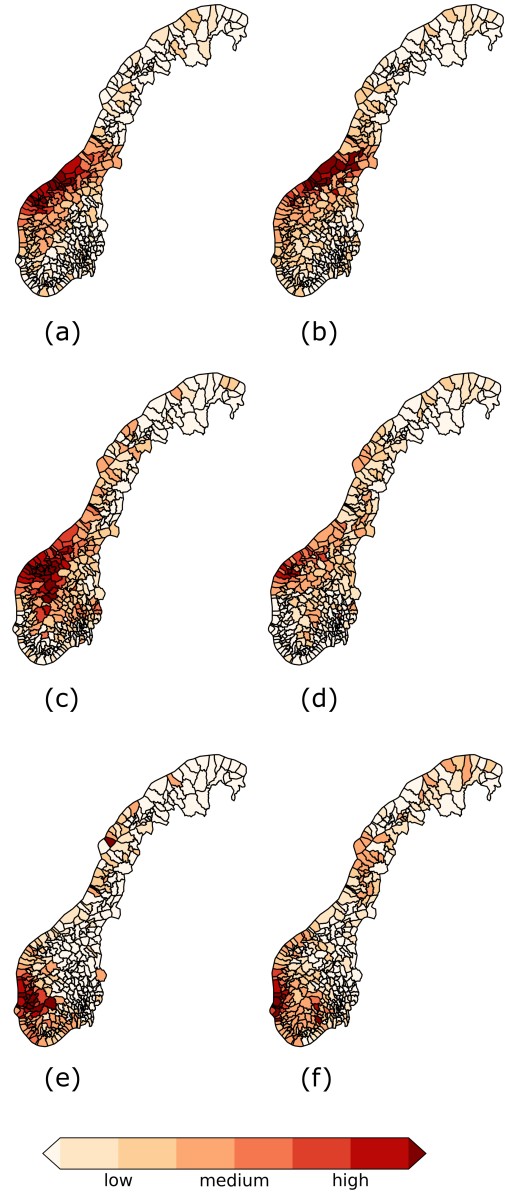

**Figure 4.** Spatial patterns of observed and estimated losses for the three most damaging events, where (a), (c) and (e) display the observed losses of the New Year storm, storm Dagmar and storm Nina, and (b), (d) and (f) are their respective estimates from the closest model to the observed loss in the testing period. The class boundaries of the colour bar are the 20th, 40th, 60th, 80th, 85th, 90th and 95th percentiles of the observed losses of their respective events. The spatial Spearman rank correlation between observed and estimated losses of the New Year storm, storm Dagmar and storm Nina are 0.67, 0.58 and 0.62 respectively. For each storm, we sum all the loss days as given in Table S1.

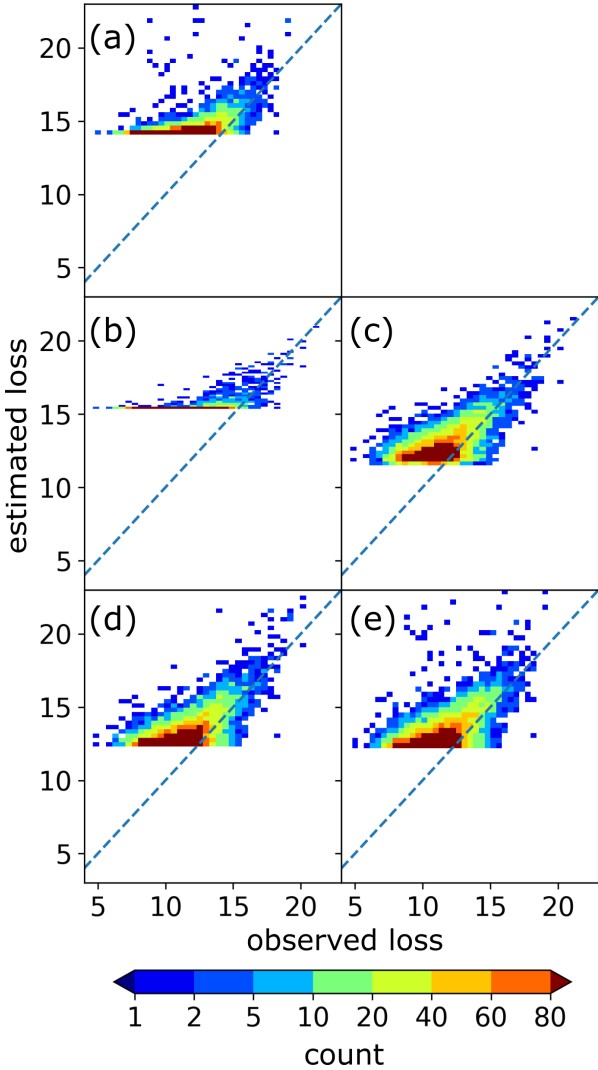

**Figure 5.** Observed and estimated daily losses (NOK) on log-log scale at the national level. Panels (a), (b), (c), (d) and (e) correspond to the ensemble mean method, the Klawa damage function, the exponential damage function, the Prahl damage function and the modified Prahl damage function, respectively. The dashed blue lines represent the 1:1 line.

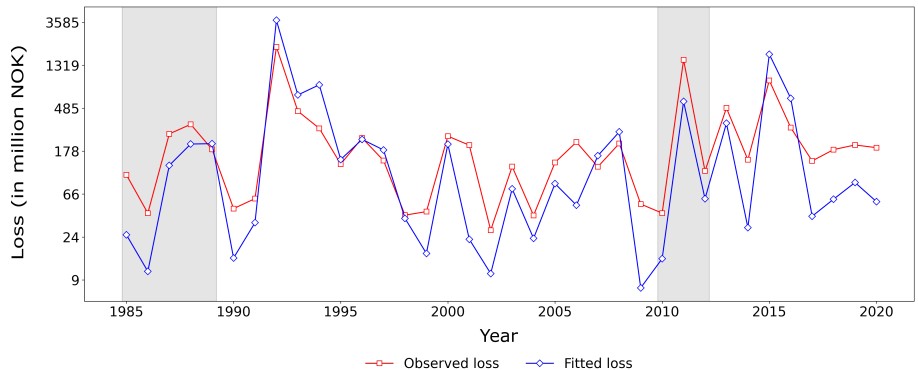

**Figure 6.** Annually-aggregated national losses using only the loss days in the extreme loss class from the insurance data (red line) along with the annual national loss estimates (blue line), which are the sum of each municipality's best-performing-model estimate (see also Table 1). Note that the y-axis is logarithmic and the shaded region represents the testing period.

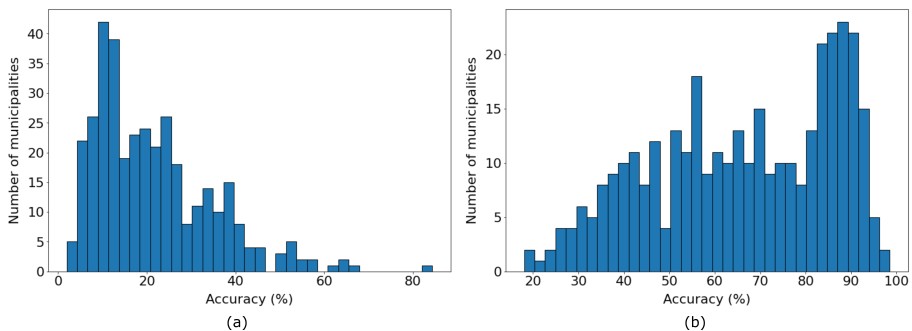

**Figure 7.** Distributions of (a) the accuracy of the damage classifier based on the wind speed 98th percentile (median: 18%) and of (b) the accuracy of the proposed damage classifier (section 2.5) devised from the Prahl damage function (median: 68%) for wind speeds above the 98th percentile, over all municipalities.

| Loss class | Damage function | Number of municipalities | MAE | CV |
|---|---|:---:|:---:|:---:|
| All loss days | Modified Prahl | 32 | 31 | 245 |
| | Prahl | 46 | 28 | 238 |
| | Klawa | 134 | 22 | 212 |
| | Exponential | 91 | 24 | 218 |
| | Ensemble mean | 53 | 26 | 226 |
| High loss class | Modified Prahl | 16 | 5 | 184 |
| | Prahl | 25 | 5 | 176 |
| | Klawa | 141 | 4 | 156 |
| | Exponential | 99 | 4 | 134 |
| | Ensemble mean | 39 | 5 | 161 |
| Extreme loss class | Modified Prahl | 53 | 73 | 153 |
| | Prahl | 65 | 67 | 147 |
| | Klawa | 121 | 49 | 132 |
| | Exponential | 75 | 55 | 143 |
| | Ensemble mean | 42 | 66 | 143 |

**Table 1.** Number of municipalities for which a model performs the best, that is has the smallest MAE as a function of the loss class, as defined in the text. The medians of the MAE and CV on all 356 municipalities are also given. Note that the results are based on the performances on the unseen testing data. Also, some municipalities are not evaluated in the high loss class due to lack of data.