# Peer review of "Assessment of wind-damage relations for Norway using 36 years of daily insurance data"

_Natural Hazards and Earth System Sciences, 2023_

## Author Comment (AC1)

**Reply to referee 2**

We thank the reviewer for reading our manuscript and giving thoughtful comments and suggestions. A detailed response to all comments is found below in blue.

The authors present a well crafted calibration exercise for storm-damage functions. They present a clear decision tree for the chosen methods and assumption. They are using a storm-based approach to statistically fit historical losses to wind speed information using different models. The basis for the calibration are high resolution insurance loss data and wind speed data covering a relatively long time period. The authors present and discuss the results in detail focusing on the high impact events and on creating a damage classifier. In the end, the authors provide a short broader discussion.

**General Comments:**

**Comment 1**

The insurance loss data and the modelled damages are obviously very skewed and mostly presented using logarithmic axis to focus on relative differenced / differences in the order of magnitude. But in the methodology, this is not incorporated as such. I would suggest the authors to change or at least expand their methodology at two points:

Thanks for the suggestions. We agree with the reviewer's point that the damages both observed and their estimates are skewed and it's not tangible to visualise it in their absolute values.

- Section 2.4.5 Ensemble mean method: As another option instead of using the arithmetic mean, I would suggest to use the mean of all the logs, as in:
  meanlog = 10 ^ ( 1/n * sum_i_n[ log( xi ) ] ) for a n loss estimates x.

  Since there are zero losses present in the loss estimates, an ensemble of the estimates with log transformation is not possible.

- Section 2.6 Model evaluation metric: I suggest to also calculate a metric that takes into account the very different order of magnitude. One option would be the mean absolute percentage error.

  We agree with the need of using different accuracy metrics. The mean absolute percentage error (MAPE) is indeed a dimensionless prediction accuracy metric.

However, the presence of zeros in loss values restricts us from using MAPE as the error metric. For the same reason, we chose to calculate the coefficient of variance, which is also a dimensionless accuracy metric that gives the dispersion of prediction around the mean.

**Comment 2**

In figure 1 the damage functions for only one municipality are shown. It is expected and written that there is a the variability of the calibrated damage functions over all municipalities, but it is not shown. It would be nice to either report the range of the calibrated parameters in a table in the supplementary material or even better reproduce a figure similar to Figure 1 showing not the points of the insurance losses but only all the calibrated damage funtions in one plot. This would provide a much better idea of the variability between the different municipalities.

We thank the reviewer for the suggestion. We have performed the proposed figure (see Fig. R1 below) and agree that it well illustrates the variability in the fits of the damage functions among the municipalities.
The fit of four damage functions analogous to Fig. 1 is shown in Fig. R1 but for all 356 municipalities. From this it is clear the fit of damage functions varies not only between models but also spatially. Figure R1a illustrates the variety of fits for the exponential damage function with very steep lines for some municipalities and much flatter lines for others. Figure R1b displays the fit for all the municipalities and highlights that the Klawa damage function doesn't increase as steeply as the other models. For the Prahl model, Fig. R1c exhibits a large variability in the fits from municipality to municipality. Figure R1e also shows that the sigmoids depicting the probability of damage occurrence have different shapes with some curves not reaching a probability of 1 within the wind speed range represented. Note that the fit can lead to negative probabilities, that we set to 0 afterwards. The shape of the magnitude term (see Eq. 8) in the Modified Prahl model can be very different among municipalities as shown in Fig. R1d with very steep or weak slopes. Therefore, we will include this figure in the supplement and some sentences on these results in the methods section.

[Figure]

Figure R1: Shapes of the damage functions for all municipalities for (a) the exponential damage function, (b) the cubic excess over threshold damage function, (c) the magnitude term in the probabilistic damage function by Prahl, and (d) the magnitude term in the modified Prahl probabilistic damage function, (e) sigmoid function that estimates the probability of an event occurrence. Note that the y-axis for (a)-(d) represents the log-loss per person with units of log NOK.

**Comment 3**

The data is split into testing and training set, which is a very important practice. I suggest to use a cross-validation approach, especially as the results in Table 1 are reported only on the unseen testing data, and municipalities have to be excluded from the evaluation due to lack of data.

We agree with the reviewer on the importance of cross-validation. Therefore, we have decided to perform a seven-fold cross-validation. The 36-year loss data is split into 7 groups in chronological order with each group containing five years of loss data (1985-1989, 1990-1994, 1995-1999, 2000-2004, 2005-2009, 2010-2014, 2015-2019) and the loss data of year 2020 is not included in any of the groups. Now taking each group as testing data, the damage functions are trained on the rest of the data. The predictive skills of the damage functions are evaluated on the testing data. The large spread in the model skill metrics (i.e., MAE and CV) indicates that the performance of damage functions is highly dependent on the choice of the training data (Fig. R2). For each model, the spread in the number of municipalities showing the smallest MAE (such as done in Table 1) remains relatively low across all loss classes, as defined in section 3.1 of the manuscript (Fig. R2 c, f, i). The black dots in Fig. R2 show the results present in the manuscript (Table 1) obtained with another set of training and testing data. We notice that they often lie outside the interquartile range, especially when considering all loss days and the extreme loss days (top and bottom rows in Fig. R2), and are sometimes even outside the range of the seven-fold cross-validation analysis, emphasising again the

strong dependence of the results to the chosen training and testing periods. In light of this new analysis, we will include these results in the manuscript and figure R2 in the supplement.

We will add some sentences in the manuscript on this topic, such as:

In section 2.6 (model evaluation section):
*The damage functions are sensitive to extreme loss observations and the presence of few extreme events can heavily alter the damage functions' shape. Therefore, different training data sets may result in differing damage function fits. Cross-validation is an effective method to estimate the uncertainties involved in the choice of the testing and training data. We perform a seven-fold cross-validation by splitting data into seven with each set of testing data having five consecutive years of data. So, in the first fold the testing period is 1985-1989 and the training period is 1990-2020, in the second fold the testing period is 1990-1994 and the remaining years are in the training dataset, and so on.*

In section 3.2:
*The seven-fold cross-validation reveals that the parameters in the storm-damage functions obtained during the fitting step depend on the choice of the training dataset. Moreover, the loss estimates are also highly dependent on the choice of training dataset (see the range in the model evaluation metrics shown in Fig. R2 a,b,d,e,g,h). However, whatever the training dataset, the Klawa and exponential models still have the best skill in most of the municipalities (as also shown in Table 1) across the different loss classes (see Fig. R2 c,f,i).*

[Figure]

Figure R2: Distribution of model performance metrics from cross-validation (a) coefficient of variance (CV), (b) mean absolute error (MAE), (c) number of municipalities with smallest MAE for four damage functions and their ensemble mean for all loss days. (d), (e) and (f) same as (a), (b) and (c) but for the high loss class. (g), (h) and (i) same as (a), (b) and (c) but for the extreme loss class. The boxes represent the interquartile range, the horizontal line represents the median, the whiskers represent the minimum and maximum and the black dots represent the results from Table 1 in the manuscript.

**Comment 4**

A short broader discussion is done in the section 4. "Conclusion". I would suggest to also discuss the following aspects:

- Discuss windspeed as explanatory value for damage model, is it able to represent the randomness of gust occurance? This is relevant for "low intensity" events, where damages are caused by infrequent and hard to predict stronger gusts. This is relevant for both damage classifier as well as estimating low intensity impacts)
- The population distribution is not changing in the chosen model setup only the total number of people. Could it be that "where people live" did not only change in scale (represented in the model) but also in location (not represented). If yes, how would this influence the model performance?
- What is the purpose of this calibration exercise, is there a foreseen application? If yes, it would be nice for the reader to know, especially what function would be the chosen

for this specific application? Even better would be a general discussion of the metrics: Which function would be the best for which type of application?

- Other readers might be needing to do a similar calibration exercise, but might lack the sample size used in this study or might face other shortcomings. It would be very interesting to discuss some learnings that could be generalized for other calibration exercises.

Thank you for the thoughtful suggestions. Here below, we discuss the four points raised by the reviewer. We will extend our discussion in the revised manuscript.

Wind speed is the most common variable used to estimate storm damages. A drawback of this approach is that the same wind speed at the municipality level resolution may cause small damages in some cases or no damages in most cases. Such inconsistencies occur mainly due to extremely local high wind gusts and incorrect reporting of damages. As a consequence, the lower end of the wind speed damage relations becomes noisy, thus making it very difficult to model. To check how the wind gust from NORA3 compares to the wind speed, we performed the same population weighting exercise with wind gusts and found a high correlation in the 98th percentiles calculated from wind speeds and wind gusts (Fig. S4b). With insurance data being at a coarser resolution than the wind gusts, which are very local (a few hundred metres), hard to predict and very transient (less than a minute), it is difficult to use wind gusts to fit damage functions.

Due to the unavailability of the gridded population data for the earlier part (1985-1999) of our study period, we had to use constant population to weight the wind speed at every grid point. Therefore, we cannot take into account the spatial change in population density, such as the spatial expansion of cities with time. This is a source of uncertainty in our storm-damage fits.

Foreseen applications can be the damage assessment right after an event, assessment of future losses in the context of climate change, and for impact-based forecasting of damages. For the future assessment, the Klawa damage function could be an obvious choice because it has an adaptation component by updating the 98th percentile with the future wind speed and it is the most common model used in the literature, which makes it easier for comparison. For the damage assessment shortly after an event and for the future assessment of losses, a set of different damage functions can be used in order to get a measure of the uncertainty in the monetary loss amount. For forecasting purposes, an ideal starting point would be to apply a damage classifier to distinguish between damaging and non-damaging winds, as part of an early warning system, followed by a prediction of losses using a variety of damage functions.

High quality data on loss and wind speed is necessary for the calibration of damage functions. A long time series of loss data is desired to reduce uncertainties and increase accuracy of model fitting and predictions. However, loss information as used in this study is rarely

available, unfortunately. In such cases, a general approach is to approximate the losses using population of the respective regions and then quantify the impact of windstorms (Donat et al., 2011). In addition, there are open source climate risk assessment models such as CLIMADA (Aznar-Siguan and Bresch 2011), which can be coupled with the loss data in hand for damage estimation.

Aznar-Siguan, G. and Bresch, D. N. (2011) CLIMADA v1: a global weather and climate risk assessment platform, Geoscientific Model Development, 12, 3085-3097, https://doi.org/10.5194/gmd-12-3085-2019

Donat, M. G., Leckebusch, G. C., Wild, S., and Ulbrich, U.: Future changes in European winter storm losses and extreme wind speeds inferred from GCM and RCM multi-model simulations, Nat. Hazards Earth Syst. Sci., 11, 1351–1370, doi:10.5194/nhess-11-1351-2011, 2011

**Specific comments:**

**Comment 1**

Title: is the phrase "in the complex terrain" justified? The only terrain specific methodology used in this paper is the usage of a population density that follows topographic features. Would "taking into account heterogenic population density" be more describing?

We agree with the reviewer that our methodology does not directly involve the Norwegian topography, which is only implicitly taken into account in the population data, with generally more people living along the sea and in the valleys than over the mountains. Therefore, following the reviewer's suggestion, we will change the manuscript title in order to better reflect our methodology, to *Assessment of wind-damage relations for Norway using 36 years of daily insurance data.*

**Comment 2**

L29: quick impact estimation for response planning directly after the event is also an important application (see Welker et al. 2021)

Thanks for mentioning this relevant point and bringing this paper to our attention. We will change the sentence to something like:

*Establishing robust windstorm-damage relations may help predict storm damage risk in the weather forecasting context (Merz et al., 2020), roughly estimate the storm impact directly after it occurred in order to better plan the emergency response (Welker et al. 2021), and evaluate the change in risk on the longer term in conjunction with climate change.*

Merz, B., Kuhlicke, C., Kunz, M., Pittore, M., Babeyko, A., Bresch, D. N., Domeisen, D. I., Feser, F., Koszalka, I., Kreibich, H., et al.: Impact forecasting to support emergency management of natural hazards, Reviews of Geophysics, 58, e2020RG000 704, 475 https://doi.org/10.1029/2020RG000704 2020.

Welker, C., Röösli, T., and Bresch, D. N.: Comparing an insurer's perspective on building damages with modelled damages from pan-European winter windstorm event sets: a case study from Zurich, Switzerland, Nat. Hazards Earth Syst. Sci., 21, 279–299, https://doi.org/10.5194/nhess-21-279-2021, 2021.

**Comment 3**

Figure 1 a) mark the threshold in the plot (compare with L170:"…,we chose  the 95th percentile of the wind speed […] as the threshold." If a modelled damage of zero would be assumed for events with a wind speed below the threshold, it would be nice to show a doted line at zero similar to Figure 1 b)

We will modify the figure as suggested by the reviewer. Although we only use the wind speed bins above the 95th percentile of the wind speed to calculate the fit, the obtained exponential model can also be applied to the wind speeds below the 95th percentile and we can get loss estimates for those wind speeds as well, as shown in Fig. 1a (red curve to the left of the 95th percentile line). However, if the estimated loss is negative, we set it to 0.

**Comment 4**

L172: "…are split into ten equally spaced bins…" I can only see 6 bins in Figure 1 a). I assume it is because 4 bins did not include the minimum of at least 5 loss days. If that is the case, I would be nice to state that for the reader. If this is not the case it would be even more important to state this.

The reviewer is right, it is because 4 bins did not include the minimum of at least 5 loss days that Fig. 1a only displays 6 bins out of the 10. We will add a sentence in line 173 to make this aspect clearer, such as: *Note that Fig. 1a only displays 6 bins because the 4 other bins do not include the minimum of 5 loss days required in each bin.*

**Comment 5**

L307ff: "The extreme loss class represents 31 and 9 ….". The structure of this sentence makes it hard to understand. Please consider a revision.

We agree and we will rephrase this sentence and better connect it with the previous sentences.

**Comment 6**

Figure 3 b) and c). Please provide a reasoning for the skewed percentiles of the non-linear class boundaries. Why are there no 5th and 10th percentile?

Figure 3a shows that most of the municipalities have a MAE between 0 and 100, whereas only a few have higher MAE. Therefore, the distribution of the MAE is highly skewed towards low MAEs and the lowest percentiles will be very similar. It is already visible in Fig. 3b, for example, with the difference between the 40th and 20th percentiles being of only 13 (28-15) in contrast to the difference between the 95th and 90th percentiles being 139 (406-267) so more than 10 times larger. Therefore, we chose to highlight the larger percentiles rather than the lower percentiles. However, we have chosen to follow the reviewer's suggestion and the updated figure is shown below:

[Figure]

Figure R3: (Left) Map of the smallest MAE among the five models in the extreme loss class fitted on the test data and (right) the corresponding coefficient of variation of the root mean square error. The legends have non-linear class boundaries at the 5th, 10th, 20th, 40th, 60th, 80th, 90th and 95th percentiles. Note that the results are based on the performances on the unseen testing data.

---

## Author Comment (AC2)

**Reply to referee 1**

We thank the reviewer for reading our manuscript and giving thoughtful comments and suggestions. A detailed response to all comments is found below in blue.

The authors assess the performance of different storm damage functions, that model the relationship between wind speeds and insured losses, for Norway. They make use of district-level insurance loss data from Norway, that span the time period 1985 to 2020, combined with regional reanalysis/hindcasts of wind speeds. The study nicely assesses the relative benefits of the different (previously developed) storm loss models.

The manuscript is overall very well written and provides a very clear overview of the topic, and insightful explanations and discussions. I only have a few minor comments for the authors to take into account before publication, and congratulate on this nice piece of work.

**Comment 1**

Line 59/60: Also the Donat et al 2011b study (already cited in the manuscript) estimated the losses at district level in Germany, regionally adjusting the loss function by Klawa&Ulbrich

Thanks for bringing this to our attention. We will add this reference in the revised manuscript.

**Comment 2**

Line 114: maybe specify that these hindcasts are "retrospective" forecasts, to avoid any ambiguity

There does not seem to be a very clear separation between a hindcast or a retrospective (reforecast) in the literature. Both usually uses a reanalysis or a state-of-the-art analysis for its boundary conditions. However a reforecast sometimes is a forecast done in hindsight using the same initial conditions as the original forecast. As the terms retrospective forecast, reforecast and hindcasts seems to be more or less interchangeably we would prefer to keep the term 'hindcast' as it is the term used in the NORA3 reference paper for the wind data we use.

**Comment 3**

Line 130-132: I note that the authors follow here a different order of operations than some of the studies they reference for the loss functions they use. E.g. the Klawa approach first calculates a loss index (i.e. the cubic exceedance of the wind speed threshold), and applies the population density weighting afterwards when spatially aggregating the losses. This is

different to what the authors are doing here, as they seem to apply the population weighting already to the daily wind speeds. It would be good to (i) be explicit about this variation in approaches, and (ii) discuss/demonstrate the effects of these different orders of operations.

The observation made by the reviewer points to an important methodological difference in our approach compared with previous studies. We chose to weight wind speeds with population first in order to have the same input for all damage functions fitting. That being said, it is also worthwhile to compare the proposed and the alternative methodology used in studies such as Pinto et al. (2007). As suggested by the reviewer, we have performed this additional analysis. The comparison of the damage functions and their predictive skill do not show any significant differences between both methodologies (Fig. R1). We have calculated the damage function (see eq. 2 in the manuscript) using both methodologies and found that there is a high correlation between the two (Fig. R1a). Upon calibrating the two damage functions with municipality level insurance losses using eq. 3, we observe that in the extreme loss class MAEs are highly correlated with each other (Fig. R1b). Moreover, we find that their magnitudes are similar, with about 91.6% of the municipalities having MAE differences within [-70 NOK, 70 NOK] (Fig. R1c). An independent sample t-test failed to conclude for any significant differences between the mean of MAEs from both methodologies. However, when not distinguishing loss classes, we find that the alternative method (such as in Pinto et al. 2007) has better skill in estimating the losses, although this result depends on the model evaluation metric used (not shown). In light of these results,  we will add some sentences in the manuscript such as:

In section 2.4.2 (Cubic-excess over threshold model):

*Here we weight the wind speeds with population and aggregate it to the municipality-level resolution such that it corresponds to the loss data resolution. However, other studies, such as Pinto et. al. (2007), weight the loss index and aggregate it to the district or national resolutions. As discussed later in the paper, these two methods do not give very different results.*

In section 3.1 (Overview of the windstorms losses):

*Unlike previous studies (e,g, Pinto et al. 2007), which weights spatially aggregated loss index devised from cubic exceedance of wind speed above sufficiently high threshold (computed as in eq. 2), we here weight the wind speed with population first and then aggregate it to a coarser resolution. We compare the Klawa damage function as in equation 2 obtained from the proposed methodology with the alternate methodology employed in Pinto et al. (2007). There is a high correlation between both damage functions/loss indices (Fig. R1a). Upon calibrating the two damage functions with municipality level insurance losses using eq. 3, we observe that in the extreme loss class MAEs are highly correlated with each other (Fig. R1b). Moreover, we find that their magnitudes are similar, with about 91.6% of the municipalities having MAE differences within [-70 NOK, 70 NOK] (Fig. R1c). An independent sample t-test*

*failed to conclude for any significant differences between the mean of MAEs from both methodologies. However, when not distinguishing loss classes, we find that the alternative method (such as in Pinto et al. 2007) has better skill in estimating the losses, although this result depends on the model evaluation metric used (not shown). Thus, there is no conclusive evidence for one of the methods exhibiting a higher predictive skill than the other.*

[Figure]

Fig R1: (a) Distribution of the Pearson correlation coefficient between the damage functions using the proposed (ours) and the alternate methodologies, (b) scatter plot of the MAEs of the extreme loss (losses above the 99.7th percentile) estimates for the proposed and the alternate methodologies using the testing data. Blue dots represents individual municipalities and the dashed red line represents the 1:1 line. (c) Distribution of MAE differences between the proposed and the alternate methodologies.

*Pinto, J. G., Fröhlich, E. L., Leckebusch, G. C., and Ulbrich, U.: Changing European storm loss potentials under modified climate conditions according to ensemble simulations of the ECHAM5/MPI-OM1 GCM, Nat. Hazards Earth Syst. Sci., 7, 165–175, https://doi.org/10.5194/nhess-7-165-2007, 2007.*

**Comment 4**

Line 146-150: I wonder how sensitive are the results to the specific choices of these testing and training samples?

The reviewer points to an important aspect of the damage functions: they are sensitive to the choice of the testing and training data (Prahl et. al. 2015). To quantify uncertainties involved in the choice of training data, we have performed a seven-fold cross-validation. The 36-year loss data is split into 7 groups in chronological order with each group containing five years of loss data (1985-1989, 1990-1994, 1995-1999, 2000-2004, 2005-2009, 2010-2014, 2015-2019) and the loss data of year 2020 is not included in any of the groups. Now taking each group as testing data, the damage functions are trained on the rest of the data. The predictive skills of the damage functions are evaluated on the testing data. The large spread in the model skill metrics (i.e., MAE and CV) indicates that the performance of damage functions is highly dependent on the choice of the training data (Fig. R2). For each model, the spread in the number of municipalities showing the smallest MAE (such as done in Table 1) remains relatively low across all loss classes, as defined in section 3.1 of the manuscript (Fig. R2 c, f, i). The black dots in Fig. R2 shows the results present in the manuscript (Table 1) obtained with another set of training and testing data. We notice that they often lie outside the

interquartile range, especially when considering all loss days and the extreme loss days (top and bottom rows in Fig. R2), and are sometimes even outside the range of the seven-fold cross-validation analysis, emphasising again the strong dependence of the results to the chosen training and testing periods. In light of this new analysis, we will include these results in the manuscript and figure R2 in the supplement.

We will add some sentences in the manuscript on this topic, such as:

In section 2.6 (model evaluation section):
*The damage functions are sensitive to extreme loss observations and the presence of few extreme events can heavily alter the damage functions' shape. Therefore, different training data sets may result in differing damage function fits. Cross-validation is an effective method to estimate the uncertainties involved in the choice of the testing and training data. We perform a seven-fold cross-validation by splitting data into seven with each set of testing data having five consecutive years of data. So, in the first fold the testing period is 1985-1989 and the training period is 1990-2020, in the second fold the testing period is 1990-1994 and the remaining years are in the training dataset, and so on.*

In section 3.2:
*The seven-fold cross-validation reveals that the parameters in the storm-damage functions obtained during the fitting step depend on the choice of the training dataset. Moreover, the loss estimates are also highly dependent on the choice of training dataset (see the range in the model evaluation metrics shown in Fig. R2 a,b,d,e,g,h). However, whatever the training dataset, the Klawa and exponential models still have the best skill in most of the municipalities (as also shown in Table 1) across the different loss classes (see Fig. R2 c,f,i).*

[Figure]

Figure R2: Distribution of model performance metrics from cross-validation (a) coefficient of variance (CV), (b) mean absolute error (MAE), (c) number of municipalities with smallest MAE for four damage functions and their ensemble mean for all loss days. (d), (e) and (f) same as (a), (b) and (c) but for the high loss class. (g), (h) and (i) same as (a), (b) and (c) but for the extreme loss class. The boxes represent the interquartile range, the horizontal line represents the median, the whiskers represent the minimum and maximum and the black dots represent the results from Table 1 in the manuscript.

Prahl, B. F., Rybski, D., Burghoff, O., and Kropp, J. P.: Comparison of storm damage functions and their performance, Nat. Hazards Earth Syst. Sci., 15, 769–788, https://doi.org/10.5194/nhess-15-769-2015, 2015.

**Comment 5**

Line 170: not clear if this statement that 82% of losses are recorded above the 95th percentile is based on the Norwegian loss data analysed against NORA3 wind speeds? Also it is not clear if it refers to 82% of loss events (as count), or 82% of loss values?

Here, 82% corresponds to the sum of the losses over all municipalities in the training period. We will adjust the sentence in the manuscript.

**Comment 6**

Line 178: In my understanding Klawa developed the function for Germany-wide losses (not districts)?

Thanks for bringing this to our attention. We will replace '*German districts*' with *'Germany'* in line 178.

**Comment 7**

Line 179: As mentioned further up, Donat et al 2011b was first calibrating the Klawa function at district level for Germany

We will rephrase line 179 to:

*Later, using the same insurance data, the damage function was calibrated by Donat et al. (2011b) for the German districts and by Pinto et al. (2012) for the affected areas of individual storm events.*

Donat, M. G., Pardowitz, T., Leckebusch, G., Ulbrich, U., and Burghoff, O.: High-resolution refinement of a storm loss model and estimation of return periods of loss-intensive storms over Germany, Natural Hazards and Earth System Sciences, 11, 2821–2833, https://doi.org/10.5194/nhess-11-2821-2011, 2011b.

Pinto, J. G., Karremann, M. K., Born, K., Della-Marta, P. M., and Klawa, M.: Loss potentials associated with European windstorms under future climate conditions, Climate Research, 54, 1–20, https://doi.org/10.3354/cr01111, 2012.

**Comment 8**

Line 182: You should specify over which time period and which seasons the percentile threshold was calculated (e.g. annual percentile, or percentile over the winter storm seasons ~October to March)?

We will rephrase line 182 to:

*This damage function takes the third power of wind speeds above the 98th percentile of the wind speed determined using the whole study period (1980-2020) scaled by the same 98th percentile [...]*

**Comment 9**

Line 197: It may be useful to clarify whether this fixed threshold of 9m/s applies to maximum gust or maximum wind speed from the NORA3 hindcast?

We will add two sentences such as:

*To alleviate this, Karremann et al. (2014b) and Little et al. (2023) suggested a 9 m/s fixed threshold for wind speed causing damage in Norway. However, in our study, we do not need this 9 m/s threshold as we use the population-weighted averaged wind speeds, reducing the relative importance of grid cells with very low wind speeds and therefore avoiding the problem of very low 98th percentile.*

Karremann, M. K., Pinto, J. G., Reyers, M., and Klawa, M.: Return periods of losses associated with European windstorm series in a changing climate, Environmental Research Letters, 9, 124 016, https://doi.org/10.1088/1748-9326/9/12/124016, 2014b.

Little, A. S., Priestley, M. D., and Catto, J. L.: Future increased risk from extratropical windstorms in northern Europe, Nature Communications, 14, 4434, https://doi.org/10.1038/s41467-023-40102-6, 2023.

**Comment 10**

Line 276: should "spread" better be "distribution"?

We agree that "distribution" is a better fit than "spread". We will change it accordingly.

**Comment 11**

Line 286: I think it is not really data-scarce, I understand that you have good/complete data but these tell you that there are only very few losses?

With 'data-scarcity', we meant that there are only very few losses and the reviewers' observation is correct in this context. We will replace 'data-scarcity' with '*rarity of loss days*'.

**Comment 12**

Line 320-325: Based on Table 1 it may be fair to say that seems to be the best function for most municipalities?

We agree with the reviewer and will add that, overall, our results suggest that the Klawa storm-damage function is the best model for a large share of municipalities (~37.6%).

**Comment 13**

Line 361: insert "is" after "This"

We will do this.

**Comment 14**

Line 390: remove the " ' " after "functions"

We will do this.

---

## Author Response (AR1)

**Reply to referee 1**

**Assessment of wind-damage relations for Norway using 36 years of daily insurance data**

We thank the reviewer for reading our manuscript and giving thoughtful comments and suggestions. A detailed response to all comments is found below in blue. The lines numbers refer to the non-tracked version of the revised manuscript.

The authors assess the performance of different storm damage functions, that model the relationship between wind speeds and insured losses, for Norway. They make use of district-level insurance loss data from Norway, that span the time period 1985 to 2020, combined with regional reanalysis/hindcasts of wind speeds. The study nicely assesses the relative benefits of the different (previously developed) storm loss models.

The manuscript is overall very well written and provides a very clear overview of the topic, and insightful explanations and discussions. I only have a few minor comments for the authors to take into account before publication, and congratulate on this nice piece of work.

**Comment 1**

Line 59/60: Also the Donat et al 2011b study (already cited in the manuscript) estimated the losses at district level in Germany, regionally adjusting the loss function by Klawa&Ulbrich

Thanks for bringing this to our attention. We have added this reference in line 57 of the revised manuscript.

*Donat et al. (2011b) estimated the losses by fitting the Klawa and Ulbrich (2003) damage function at district level for Germany.*

**Comment 2**

Line 114: maybe specify that these hindcasts are "retrospective" forecasts, to avoid any ambiguity

There does not seem to be a very clear separation between a hindcast or a retrospective (reforecast) in the literature. Both usually uses a reanalysis or a state-of-the-art analysis for its boundary conditions. However a reforecast sometimes is a forecast done in hindsight using the same initial conditions as the original forecast. As the terms retrospective forecast, reforecast and hindcasts seem to be more or less interchangeably, we prefer to keep the term 'hindcast' as it is the term used in the NORA3 reference paper for the wind data we use.

**Comment 3**

Line 130-132: I note that the authors follow here a different order of operations than some of the studies they reference for the loss functions they use. E.g. the Klawa approach first calculates a loss index (i.e. the cubic exceedance of the wind speed threshold), and applies the population density weighting afterwards when spatially aggregating the losses. This is different to what the authors are doing here, as they seem to apply the population weighting already to the daily wind speeds. It would be good to (i) be explicit about this variation in approaches, and (ii) discuss/demonstrate the effects of these different orders of operations.

The observation made by the reviewer points to an important methodological difference in our approach compared with previous studies. We chose to weight wind speeds with population first in order to have the same input for all damage functions fitting. That being said, it is also worthwhile to compare the proposed and the alternative methodology used in studies such as Pinto et al. (2007). As suggested by the reviewer, we have performed this additional analysis. The comparison of the damage functions and their predictive skill do not show any significant differences between both methodologies (Fig. R1). We have calculated the damage function (see eq. 2 in the manuscript) using both methodologies and found that there is a high correlation between the two (Fig. R1a). Upon calibrating the two damage functions with municipality level insurance losses using eq. 3, we observe that in the extreme loss class MAEs are highly correlated with each other (Fig. R1b). Moreover, we find that their magnitudes are similar, with about 91.6% of the municipalities having MAE differences within [-70, 70] NOK/person in the testing data (Fig. R1c). An independent sample t-test failed to conclude for any significant differences between the mean of MAEs from both methodologies. However, when not distinguishing loss classes, we find that the alternative method (such as in Pinto et al. 2007) has better skill in estimating the losses, although this result depends on the model evaluation metric used (not shown). In light of these results, we have added Fig. R1 and some text to the supplement (Fig. S7) as well as the following sentences in the revised manuscript:

In section 2.4.2 (Cubic-excess over threshold model), lines 209-211:

*Here we weight the wind speeds with population and aggregate it to the municipality-level resolution such that it corresponds to the loss data resolution. However, other studies, such as Pinto et. al. (2007), weight the loss index and aggregate it to the district or national resolutions. As discussed later in the paper, these two methods do not give very different results.*

In section 3.2 (Municipality level loss estimations), lines 354-360:

*Unlike previous studies (e.g., Pinto et al., 2007), which weighted the spatially aggregated loss index devised from cubic exceedance of wind speed above a sufficiently high threshold (computed as in eq. 2), we here weight the wind speed with population first and then*

*aggregate it to a coarser resolution. We compare the Klawa damage function as in eq. 2 obtained from the proposed methodology with the alternative methodology employed in Pinto et al. (2007). We found both the damage estimates and its error to be strongly correlated (Fig. S7a). An independent sample t-test failed to conclude for any significant differences between the mean of MAEs from both methodologies. A detailed comparison can be found in the supplemental material.*

[Figure]

Fig R1: (a) Distribution of the Pearson correlation coefficient between the damage functions using the proposed (ours) and the alternate methodologies, (b) scatter plot of the MAEs of the extreme loss (losses above the 99.7th percentile) estimates for the proposed and the alternate methodologies using the testing data. Blue dots represent individual municipalities and the dashed red line represents the 1:1 line. (c) Distribution of MAE differences between the proposed and the alternate methodologies.

*Pinto, J. G., Fröhlich, E. L., Leckebusch, G. C., and Ulbrich, U.: Changing European storm loss potentials under modified climate conditions according to ensemble simulations of the ECHAM5/MPI-OM1 GCM, Nat. Hazards Earth Syst. Sci., 7, 165–175, https://doi.org/10.5194/nhess-7-165-2007, 2007.*

**Comment 4**

Line 146-150: I wonder how sensitive are the results to the specific choices of these testing and training samples?

The reviewer points to an important aspect of the damage functions: they are sensitive to the choice of the testing and training data (Prahl et. al. 2015). To quantify uncertainties involved in the choice of training data, we have performed a seven fold cross-validation. The 36-year loss data is splitted into 7 groups in chronological order with each group containing five years of loss data (1985-1989, 1990-1994, 1995-1999, 2000-2004, 2005-2009, 2010-2014, 2015-2019) and the loss data of year 2020 is not included in any of the groups. Now taking each group as testing data, the damage functions are trained on the rest of the data. The predictive skills of the damage functions are evaluated on the testing data. The large spread in the model skill metrics (i.e., MAE and CV) indicates that the performance of damage functions is highly dependent on the choice of the training data (Fig. R2). For each model, the spread in the number of municipalities showing the smallest MAE (such as done in Table 1) remains relatively low across all loss classes, as defined in section 3.1 of the manuscript (Fig. R2 c, f, i). The black dots in Fig. R2 shows the results present in the manuscript (Table 1) obtained

with another set of training and testing data. We notice that they often lie outside the interquartile range, especially when considering all loss days and the extreme loss days (top and bottom rows in Fig. R2), and are sometimes even outside the range of the seven-fold cross-validation analysis, emphasising again the strong dependence of the results to the chosen training and testing periods. In light of this new analysis, we have included these results in the manuscript and the figure R2 in the supplement (Fig. S8).

We have added the following sentences in the manuscript on this topic:

In section 2.6 (model evaluation section), lines 285-290:
*The damage functions are sensitive to extreme loss observations and the presence of few extreme events can heavily alter the damage functions' shape. Therefore, different training data sets may result in differing damage function fits. Cross-validation is an effective method to estimate the uncertainties involved in the choice of the testing and training data. We perform a seven fold cross-validation by splitting data into seven with each set of testing data having five consecutive years of data. So, in the first fold the testing period is 1985-1989 and the training period is 1990-2020, in the second fold the testing period is 1990-1994 and the remaining years are in the training dataset, and so on.*

In section 3.2 (Municipality level loss estimations), lines 361-365:
*The seven fold cross-validation reveals that the parameters in the storm-damage functions obtained during the fitting step depend on the choice of the training dataset. Moreover, the model evaluation metrics are highly dependent on the choice of the training dataset (see the range in Fig. S8 a,b,d,e,g,h). However, independent of the training dataset, the Klawa and exponential models have the best skill in most of the municipalities (as also shown in Table 1) across the different loss classes (see Fig. S8 c,f,i).*

[Figure]

Figure R2: Distribution of model performance metrics from cross-validation (a) coefficient of variance (CV), (b) mean absolute error (MAE), (c) number of municipalities with smallest MAE for four damage functions and their ensemble mean for all loss days. (d), (e) and (f) same as (a), (b) and (c) but for the high loss class. (g), (h) and (i) same as (a), (b) and (c) but for the extreme loss class. The boxes represent the interquartile range, the horizontal line represents the median, the whiskers represent the minimum and maximum and the black dots represent the results from Table 1 in the manuscript.

Prahl, B. F., Rybski, D., Burghoff, O., and Kropp, J. P.: Comparison of storm damage functions and their performance, Nat. Hazards Earth Syst. Sci., 15, 769–788, https://doi.org/10.5194/nhess-15-769-2015, 2015.

**Comment 5**

Line 170: not clear if this statement that 82% of losses are recorded above the 95th percentile is based on the Norwegian loss data analysed against NORA3 wind speeds? Also it is not clear if it refers to 82% of loss events (as count), or 82% of loss values?

Here, 82% corresponds to the sum of the losses over all municipalities in the training period. We adjusted the sentence in the line 172 of the revised manuscript.

*To take advantage of this, we choose the 95th percentile of the population weighted wind speed in each municipality as the threshold for the exponential model above which the aggregated losses represent 82% of the national losses that occurred in the training period.*

**Comment 6**

Line 178: In my understanding Klawa developed the function for Germany-wide losses (not districts)?

Thanks for bringing this to our attention. We have replaced '*German districts*' with *'Germany'* in line 184 of the revised manuscript.

**Comment 7**

Line 179: As mentioned further up, Donat et al 2011b was first calibrating the Klawa function at district level for Germany

We rephrase this in line 185 of the revised manuscript:

*Later, using the same insurance data, the damage function was calibrated by Donat et al. (2011b) for the German districts and by Pinto et al. (2012) for the affected areas of individual storm events.*

Donat, M. G., Pardowitz, T., Leckebusch, G., Ulbrich, U., and Burghoff, O.: High-resolution refinement of a storm loss model and estimation of return periods of loss-intensive storms over Germany, Natural Hazards and Earth System Sciences, 11, 2821–2833, https://doi.org/10.5194/nhess-11-2821-2011, 2011b.

Pinto, J. G., Karremann, M. K., Born, K., Della-Marta, P. M., and Klawa, M.: Loss potentials associated with European windstorms under future climate conditions, Climate Research, 54, 1–20, https://doi.org/10.3354/cr01111, 2012.

**Comment 8**

Line 182: You should specify over which time period and which seasons the percentile threshold was calculated (e.g. annual percentile, or percentile over the winter storm seasons ~October to March)?

We rephrase this in line 188 of the revised manuscript:

*This damage function takes the third power of wind speeds above the 98th percentile of the wind speed determined using the whole study period (1980-2020) scaled by the same 98th percentile [...]*

**Comment 9**

Line 197: It may be useful to clarify whether this fixed threshold of 9m/s applies to maximum gust or maximum wind speed from the NORA3 hindcast?

We have made this clearer in line 205 of the revised manuscript:

*However, in our study, we do not need this 9 m/s threshold as we use the population-weighted averaged wind speeds, reducing the relative importance of grid cells with very low wind speeds and therefore avoiding the problem of very low 98th percentile.*

**Comment 10**

Line 276: should "spread" better be "distribution"?

We agree that "distribution" is a better fit than "spread". This change is made in line 291 of the revised manuscript.

**Comment 11**

Line 286: I think it is not really data-scarce, I understand that you have good/complete data but these tell you that there are only very few losses?

With 'data-scarcity', we meant that there are only very few losses and the reviewers' observation is correct in this context. We have replaced 'data-scarcity' with '*rarity of loss days*' in line 302 of the revised manuscript.

**Comment 12**

Line 320-325: Based on Table 1 it may be fair to say that seems to be the best function for most municipalities?

We agree with the reviewer and in line 341 we have added:

*Overall, our results suggest that the Klawa storm-damage function is the best model for a large share of municipalities (37.6%).*

**Comment 13**

Line 361: insert "is" after "This"

Done (line 397 in the revised manuscript).

**Comment 14**

Line 390: remove the " ' " after "functions"

Done (line 426 in the revised manuscript).

**Reply to referee 2**

**Assessment of wind-damage relations for Norway using 36 years of daily insurance data**

We thank the reviewer for reading our manuscript and giving thoughtful comments and suggestions. A detailed response to all comments is found below in blue. The lines numbers refer to the non-tracked version of the revised manuscript.

The authors present a well crafted calibration exercise for storm-damage functions. They present a clear decision tree for the chosen methods and assumption. They are using a storm-based approach to statistically fit historical losses to wind speed information using different models. The basis for the calibration are high resolution insurance loss data and wind speed data covering a relatively long time period. The authors present and discuss the results in detail focusing on the high impact events and on creating a damage classifier. In the end, the authors provide a short broader discussion.

**General Comments:**

**Comment 1**

The insurance loss data and the modelled damages are obviously very skewed and mostly presented using logarithmic axis to focus on relative differenced / differences in the order of magnitude. But in the methodology, this is not incorporated as such. I would suggest the authors to change or at least expand their methodology at two points:

Thank you for the suggestions. We agree with the reviewer's point that the damages both observed and their estimates are skewed and it is not tangible to visualise it in their absolute values.

- Section 2.4.5 Ensemble mean method: As another option instead of using the arithmetic mean, I would suggest to use the mean of all the logs, as in:
  meanlog $= 10 \wedge ( 1/n * \text{sum\_i\_n}[ \log( xi ) ] )$ for a n loss estimates x.

  Since there are zero losses present in the loss estimates, an ensemble of the estimates with log transformation is not possible.

- Section 2.6 Model evaluation metric: I suggest to also calculate a metric that takes into account the very different order of magnitude. One option would be the mean absolute percentage error.

We agree with the need of using different accuracy metrics. The mean absolute percentage error (MAPE) is indeed a dimensionless prediction accuracy metric. However, the presence of zeros in loss values restricts us from using MAPE as the error metric. For the same reason, we chose to calculate the coefficient of variance, which is also a dimensionless accuracy metric that gives the dispersion of prediction around the mean.

**Comment 2**

In figure 1 the damage functions for only one municipality are shown. It is expected and written that there is a the variability of the calibrated damage functions over all municipalities, but it is not shown. It would be nice to either report the range of the calibrated parameters in a table in the supplementary material or even better reproduce a figure similar to Figure 1 showing not the points of the insurance losses but only all the calibrated damage funtions in one plot. This would provide a much better idea of the variability between the different municipalities.

We thank the reviewer for the suggestion. We have performed the proposed figure (see Fig. R1 below) and agree that it well illustrates the variability in the fits of the damage functions among the municipalities.

The fit of four damage functions analogous to Fig. 1 is shown in Fig. R1 but for all 356 municipalities. From this it is clear the fit of damage functions varies not only between models but also spatially. Figure R1a illustrates the variety of fits for the exponential damage function with very steep lines for some municipalities and much flatter lines for others. Figure R1b displays the fit for all the municipalities and highlights that the Klawa damage function doesn't increase as steeply as the other models. For the Prahl model, Fig. R1c exhibits a large variability in the fits from municipality to municipality. Figure R1e also shows that the sigmoids depicting the probability of damage occurrence have different shapes with some curves not reaching a probability of 1 within the wind speed range represented. Note that the fit can lead to negative probabilities, that we set to 0 afterwards. The shape of the magnitude term (see Eq. 8) in the Modified Prahl model can be very different among municipalities as shown in Fig. R1d with very steep or weak slopes. Therefore, we have included this figure in the supplement and some sentences on these results in the methods section.

In the revised manuscript, we have added the following text in lines 366-372 in section 3.2 (Municipality level loss estimations):

*The fits of the four damage functions and of the probability term largely vary not only between models but also from municipality to municipality (Fig. S9). Figure S9a,c,d illustrates the variety among the fits for the exponential, Prahl and modified Prahl damage functions with very steep lines for some municipalities and much flatter lines for others. Figure S9b highlights that the Klawa damage function does not increase as steeply as the other models and the*

*variability among the municipalities is smaller. Finally, Fig. S9e also shows that the sigmoids depicting the probability of damage occurrence have different shapes in different municipalities with some curves not reaching a loss probability of 1 within the wind speed range represented. Note that the fit can lead to negative probabilities, that we set to 0 afterwards.*

[Figure]

Figure R1: Shapes of the damage functions for all municipalities for (a) the exponential damage function, (b) the cubic excess over threshold damage function, (c) the magnitude term in the probabilistic damage function by Prahl, and (d) the magnitude term in the modified Prahl probabilistic damage function, (e) sigmoid function that estimates the probability of an event occurrence. Note that the y-axis for (a)-(d) represents the log-loss per person with units of log NOK.

**Comment 3**

The data is split into testing and training set, which is a very important practice. I suggest to use a cross-validation approach, especially as the results in Table 1 are reported only on the unseen testing data, and municipalities have to be excluded from the evaluation due to lack of data.

The reviewer points to an important aspect of the damage functions: they are sensitive to the choice of the testing and training data (Prahl et. al. 2015). To quantify uncertainties involved in the choice of training data, we have performed a seven fold cross-validation. The 36-year loss data is splitted into 7 groups in chronological order with each group containing five years of loss data (1985-1989, 1990-1994, 1995-1999, 2000-2004, 2005-2009, 2010-2014, 2015-2019) and the loss data of year 2020 is not included in any of the groups. Now taking each group as testing data, the damage functions are trained on the rest of the data. The predictive skills of the damage functions are evaluated on the testing data. The large spread in the model skill metrics (i.e., MAE and CV) indicates that the performance of damage functions is

highly dependent on the choice of the training data (Fig. R2). For each model, the spread in the number of municipalities showing the smallest MAE (such as done in Table 1) remains relatively low across all loss classes, as defined in section 3.1 of the manuscript (Fig. R2 c, f, i). The black dots in Fig. R2 shows the results present in the manuscript (Table 1) obtained with another set of training and testing data. We notice that they often lie outside the interquartile range, especially when considering all loss days and the extreme loss days (top and bottom rows in Fig. R2), and are sometimes even outside the range of the seven-fold cross-validation analysis, emphasising again the strong dependence of the results to the chosen training and testing periods. In light of this new analysis, we have included these results in the manuscript and the figure R2 in the supplement (Fig. S8).

We have added the following sentences in the manuscript on this topic:

In section 2.6 (model evaluation section), lines 285-290:
*The damage functions are sensitive to extreme loss observations and the presence of few extreme events can heavily alter the damage functions' shape. Therefore, different training data sets may result in differing damage function fits. Cross-validation is an effective method to estimate the uncertainties involved in the choice of the testing and training data. We perform a seven fold cross-validation by splitting data into seven with each set of testing data having five consecutive years of data. So, in the first fold the testing period is 1985-1989 and the training period is 1990-2020, in the second fold the testing period is 1990-1994 and the remaining years are in the training dataset, and so on.*

In section 3.2 (Municipality level loss estimations), lines 361-365:
*The seven fold cross-validation reveals that the parameters in the storm-damage functions obtained during the fitting step depend on the choice of the training dataset. Moreover, the model evaluation metrics are highly dependent on the choice of the training dataset (see the range in Fig. S8 a,b,d,e,g,h). However, independent of the training dataset, the Klawa and exponential models have the best skill in most of the municipalities (as also shown in Table 1) across the different loss classes (see Fig. S8 c,f,i).*

[Figure]

Figure R2: Distribution of model performance metrics from cross-validation (a) coefficient of variance (CV), (b) mean absolute error (MAE), (c) number of municipalities with smallest MAE for four damage functions and their ensemble mean for all loss days. (d), (e) and (f) same as (a), (b) and (c) but for the high loss class. (g), (h) and (i) same as (a), (b) and (c) but for the extreme loss class. The boxes represent the interquartile range, the horizontal line represents the median, the whiskers represent the minimum and maximum and the black dots represent the results from Table 1 in the manuscript.

**Comment 4**

A short broader discussion is done in the section 4. "Conclusion". I would suggest to also discuss the following aspects:

- Discuss windspeed as explanatory value for damage model, is it able to represent the randomness of gust occurance? This is relevant for "low intensity" events, where damages are caused by infrequent and hard to predict stronger gusts. This is relevant for both damage classifier as well as estimating low intensity impacts)
- The population distribution is not changing in the chosen model setup only the total number of people. Could it be that "where people live" did not only change in scale (represented in the model) but also in location (not represented). If yes, how would this influence the model performance?
- What is the purpose of this calibration exercise, is there a foreseen application? If yes, it would be nice for the reader to know, especially what function would be the chosen

for this specific application? Even better would be a general discussion of the metrics: Which function would be the best for which type of application?

- Other readers might be needing to do a similar calibration exercise, but might lack the sample size used in this study or might face other shortcomings. It would be very interesting to discuss some learnings that could be generalized for other calibration exercises.

Thank you for the thoughtful suggestions. Here below, we discuss the four points raised by the reviewer. We have expanded our discussion in the revised manuscript.

In lines 432-457 we added,

*Wind speed is the most common variable used to estimate storm damages. A drawback of this approach is that the same wind speed at the municipality level resolution may cause small damages in some cases or no damages in most cases. Such inconsistencies occur mainly due to extremely local high wind gusts and incorrect reporting of damages. As a consequence, the lower end of the wind speed-damage relations becomes noisy, thus making it very difficult to model. To check how the wind gust from NORA3 compares to the wind speed, we performed the same population weighting exercise with daily maximum wind gusts and found a high correlation (0.91) in the 98th percentiles calculated from wind speeds and wind gusts (Fig. S4b). With insurance data being at a coarser resolution than the wind gusts, which are very local (a few hundred metres), it is difficult to get meaningful wind gust values at municipality level because the impact of high values will be weakened by the population-weighted averaging step.*

*Due to the unavailability of the gridded population data for the earlier part (1985-1999) of our study period, we had to use a constant spatial distribution of the population to weight the wind speed at every grid point. Therefore, we cannot take into account the spatial change in population density, such as the spatial expansion of cities with time. This is a source of uncertainty in our storm-damage fits.*

*High quality data on loss and wind speed is necessary for the calibration of damage functions. Long time series of loss data are desired to reduce uncertainties and increase accuracy of model fitting and predictions. However, loss information as used in this study is rarely available. In such cases, a general approach is to approximate the losses using population of the respective regions and then quantify the impact of windstorms (Donat et al., 2011a). In addition, there are open source climate risk assessment models such as CLIMADA (Aznar-Siguan and Bresch, 2019), which can be coupled with loss data for damage estimation.*

*There are several limitations to the damage functions including the inability of the models to account for the duration of the events and their tuning to model extreme losses at the expense of the low losses. Furthermore, the randomness of losses towards the lower loss spectrum*

*diminishes the damage classifier's predictive skill. There are also certain pitfalls in the insurance data, such as incorrect reporting of time, location and type of claims. Also, the slight underestimation of maximum wind speeds in NORA3 may affect the shape of the damage curves. A direct comparison between other studies that employ damage functions is not possible because the unit of loss in this study is NOK per person while most other studies use the loss ratio (insured loss/total value of the insured assets) instead of the actual insured loss.*

In lines 458-460:
*Applications of damage functions can range from impact-based forecasting of damage, to damage assessment right after an event, as well as assessment of future losses in the context of climate change with an ensemble of wind-damage relations providing a measure of the uncertainty in the monetary loss amount.*

In lines 467-469:
*For forecasting purposes, an ideal starting point would be to apply a damage classifier to distinguish between damaging and non-damaging winds, as part of an early warning system, followed by a prediction of losses using a variety of damage functions.*

Aznar-Siguan, G. and Bresch, D. N. (2011) CLIMADA v1: a global weather and climate risk assessment platform, Geoscientific Model Development, 12, 3085-3097, https://doi.org/10.5194/gmd-12-3085-2019

Donat, M. G., Leckebusch, G. C., Wild, S., and Ulbrich, U.: Future changes in European winter storm losses and extreme wind speeds inferred from GCM and RCM multi-model simulations, Nat. Hazards Earth Syst. Sci., 11, 1351–1370, doi:10.5194/nhess-11-1351-2011, 2011

**Specific comments:**

**Comment 1**

Title: is the phrase "in the complex terrain" justified? The only terrain specific methodology used in this paper is the usage of a population density that follows topographic features. Would "taking into account heterogenic population density" be more describing?

We agree with the reviewer that our methodology does not directly involve the Norwegian topography, which is only implicitly taken into account in the population data, with generally more people living along the sea and in the valleys than over the mountains. Therefore, following the reviewer's suggestion, we have changed the manuscript title in order to better reflect our methodology, to *Assessment of wind-damage relations for Norway using 36 years of daily insurance data.*

**Comment 2**

L29: quick impact estimation for response planning directly after the event is also an important application (see Welker et al. 2021)

Thanks for mentioning this relevant point and bringing this paper to our attention. We have added this reference in line 28 of the revised manuscript as:

*Establishing robust windstorm-damage relations may help predict storm damage risk in the weather forecasting context (Merz et al., 2020), roughly estimate the storm impact directly after it occurred in order to better plan the emergency response (Welker et al. 2021), and evaluate the change in risk on the longer term in conjunction with climate change.*

Merz, B., Kuhlicke, C., Kunz, M., Pittore, M., Babeyko, A., Bresch, D. N., Domeisen, D. I., Feser, F., Koszalka, I., Kreibich, H., et al.: Impact forecasting to support emergency management of natural hazards, Reviews of Geophysics, 58, e2020RG000 704, 475 https://doi.org/10.1029/2020RG000704 2020.

Welker, C., Röösli, T., and Bresch, D. N.: Comparing an insurer's perspective on building damages with modelled damages from pan-European winter windstorm event sets: a case study from Zurich, Switzerland, Nat. Hazards Earth Syst. Sci., 21, 279–299, https://doi.org/10.5194/nhess-21-279-2021, 2021.

**Comment 3**

Figure 1 a) mark the threshold in the plot (compare with L170:"…,we chose the 95th percentile of the wind speed […] as the threshold." If a modelled damage of zero would be assumed for events with a wind speed below the threshold, it would be nice to show a doted line at zero similar to Figure 1 b)

We have modified the figure as suggested by the reviewer. Although we only use the wind speed bins above the 95th percentile of the wind speed to calculate the fit, the obtained exponential model can also be applied to the wind speeds below the 95th percentile and we can get loss estimates for those wind speeds as well, as shown in Fig. 1a (red curve to the left of the 95th percentile line). In the revised manuscript, we have made this clear in line 179:

*Although we only use the wind speed bins above the 95th percentile of the wind speed to calculate the fit, the obtained exponential model can also be applied to the wind speeds below the 95th percentile and we can get loss estimates for those wind speeds as well, as shown in Fig. 1a (see the red dashed line).*

**Comment 4**

L172: "…are split into ten equally spaced bins…" I can only see 6 bins in Figure 1 a). I assume it is because 4 bins did not include the minimum of at least 5 loss days. If that is the case, I would be nice to state that for the reader. If this is not the case it would be even more important to state this.

The reviewer is right, it is because 4 bins did not include the minimum of at least 5 loss days that Fig. 1a only displays 6 bins out of the 10. We added the following sentence in line 176 of the revised manuscript to make this aspect clearer,

*Note that Fig. 1a only displays 6 bins because the 4 other bins do not include the minimum of 5 loss days required in each bin.*

**Comment 5**

L307ff: "The extreme loss class represents 31 and 9 …." The structure of this sentence makes it hard to understand. Please consider a revision.

We agree and we rephrase this sentence in lines 322-324 of the revised manuscript to better connect it with the previous sentences.

*The aggregated municipality losses in the extreme loss class account for 85% of the total national loss, while the high loss class comprises 8% of the total national loss. In each municipality, the extreme loss class includes approximately 31 days in the training data and 9 days in the testing data (occurring on average around once a year).*

**Comment 6**

Figure 3 b) and c). Please provide a reasoning for the skewed percentiles of the non-linear class boundaries. Why are there no 5th and 10th percentile?

Figure 3a shows that most of the municipalities have a MAE between 0 and 100, whereas only a few have higher MAE. Therefore, the distribution of the MAE is highly skewed towards low MAEs and the lowest percentiles will be very similar. It is already visible in Fig. 3b, for example, with the difference between the 40th and 20th percentiles being of only 13 (28-15) in contrast to the difference between the 95th and 90th percentiles being 139 (406-267) so more than 10 times larger. Therefore, we chose to highlight the larger percentiles rather than the lower percentiles. However, we have chosen to follow the reviewer's suggestion and the updated figure is shown below:

[Figure]

Figure R3: (Left) Map of the smallest MAE among the five models in the extreme loss class fitted on the test data and (right) the corresponding coefficient of variation of the root mean square error. The legends have non-linear class boundaries at the 5th, 10th, 20th, 40th, 60th, 80th, 90th and 95th percentiles. Note that the results are based on the performances on the unseen testing data.